# Excess *S*-adenosylmethionine inhibits methylation via catabolism to adenine

Kazuki Fukumoto[1,6,9], Kakeru Ito[1,9], Benjamin Saer[2,9], George Taylor[3], Shiqi Ye[1,7], Mayu Yamano [1], Yuki Toriba [1,8], Andrew Hayes [4], Hitoshi Okamura [5✉] & Jean-Michel Fustin [1,2✉]

The global dietary supplement market is valued at over USD 100 billion. One popular dietary supplement, *S*-adenosylmethionine, is marketed to improve joints, liver health and emotional well-being in the US since 1999, and has been a prescription drug in Europe to treat depression and arthritis since 1975, but recent studies questioned its efficacy. In our body, *S*-adenosylmethionine is critical for the methylation of nucleic acids, proteins and many other targets. The marketing of SAM implies that more *S*-adenosylmethionine is better since it would stimulate methylations and improve health. Previously, we have shown that methylation reactions regulate biological rhythms in many organisms. Here, using biological rhythms to assess the effects of exogenous *S*-adenosylmethionine, we reveal that excess *S*-adenosylmethionine disrupts rhythms and, rather than promoting methylation, is catabolized to adenine and methylthioadenosine, toxic methylation inhibitors. These findings further our understanding of methyl metabolism and question the safety of *S*-adenosylmethionine as a supplement.

[1] Kyoto University, Graduate School of Pharmaceutical Sciences, Department of Molecular Metabology, Kyoto, Japan. [2] The University of Manchester, Centre for Biological Timing, Manchester, UK. [3] The University of Manchester, BioMS Core Facility, Manchester, UK. [4] The University of Manchester, Genomics Technologies Core Facility, Manchester, UK. [5] Kyoto University, Graduate School of Medicine, Division of Physiology and Neurobiology, Kyoto, Japan. [6]Present address: Kokando Pharmaceutical Co., Ltd, Tokyo, Japan. [7]Present address: Cancer Epigenetics Laboratory, Francis Crick Institute, Cambridge, UK. [8]Present address: Master's Programme in Molecular Biology, Faculty of Science and Bioengineering Sciences, Vrije Universiteit Brussel, Brussels, Belgium. [9]These authors contributed equally: Kazuki Fukumoto, Kakeru Ito, Benjamin Saer. ✉email: okamura.hitoshi.4u@kyoto-u.ac.jp; jean-michel.fustin@manchester.ac.uk

S-Adenosylmethionine (SAM) is the second-most used enzyme substrate, after ATP[1]. It is used by over 200 methyltransferase enzymes (MTases) in human to methylate histone and non-histone proteins, nucleic acids (DNA and RNA), phospholipids, hormones, and small molecules. Methyl metabolism is ancient and well conserved from bacteria to humans. SAM is synthesized from methionine and ATP, and subsequently used in reactions catalyzed by MTases, leading to the release of S-adenosylhomocysteine (SAH). SAH is a competitive inhibitor of MTases and is hydrolyzed by *adenosylhomocysteinase* (AHCY) to homocysteine, the latter re-methylated to methionine using methyltetrahydrofolate or betaine depending on the biological system[2]. This pathway is commonly referred to as the methyl cycle.

Abnormal methyl cycle and dependent methylations can be caused by genetic alteration, toxic chemicals, or dietary deficiencies[3], and contribute to the etiology of many pathologies including cancer[4,5], diabetes[6], atherosclerosis and cardiovascular diseases[7–9], birth defects[10], and neurodegenerative diseases[11].

While the methyl cycle requires the essential nutrients methionine, folic acid (for $CH_3$-THF), and choline (as a precursor of betaine), SAM is itself not considered one. SAM is, however, available as a nutritional supplement (manufactured as SAMe or SAM-e) to promote emotional well-being and healthy joints, at a recommended dose of 0.8–1.6 g/day. The use of SAM for the treatment of depression originates from reports showing that severely depressed patients and patients with Alzheimer's dementia had lower SAM levels in the cerebrospinal fluid[12], highlighting a link between depression and methyl metabolism[13]. A meta-analysis of clinical studies investigating SAM for the treatment of depression, osteoarthritis and liver diseases concluded that SAM was associated with some improvement compared to placebo, and called for further studies[14]. In 2010, a double-blind, randomized, and placebo-controlled clinical trial sought to further investigate SAM in the treatment of depression, but reported no improvements over placebo[15]. A heated debate on the use of SAM followed[16,17]. A later report suggested that SAM might be effective, but only in males[18]. In 2020, the same team further reported that a higher dose (3200 mg/day) failed to improve depression beyond placebo effects, but side effects including abdominal discomfort (31%) and fluid retention (25%) became significantly more common[19]. Similarly, a meta-analysis of studies using SAM for the treatment of chronic liver diseases concluded SAM[20] had limited clinical values. SAM appears to have some benefits for the management of osteoarthritis symptoms, but the mechanisms are unknown[21]. Importantly, the consequences of chronic oral SAM administration on methyl metabolism are unknown, and the few existing studies have only looked at the immediate effects (<24 h) of SAM on human plasma methyl metabolites[22].

Notably due to most studies failing to highlight SAM as an efficient therapy, and its unknown modes of action to this day, SAM is not often proposed as a treatment. Despite this, SAM, as a food supplement, can be sold over the counter in many countries without clear evidence of its benefits or safety. Scientific evidence, however, clearly indicates that excesses in metabolites related to methyl metabolism such as vitamin B9 (folic acid) and methionine can be as detrimental as deficiencies[8,23], which raises serious questions regarding SAM. Indeed, in contrast with these unsubstantiated benefits of SAM, SAM has clearly been demonstrated to have anti-proliferative, pro-apoptotic, and anti-metastatic effects in cancer cell lines, although the mechanisms were not identified[24–26]. While this may be of considerable value in cancer chemotherapy, the induction of cell cycle arrest further indicates SAM may potentially be toxic. Interestingly, in liver cancer cells SAM was shown to cause genome-wide hypomethylation as well as hypermethylation of DNA depending on the locus, indicating that more SAM does not necessarily mean more methylations[26].

We have previously reported that circadian rhythms, i.e., biological rhythms ticking with a period close to 24 h, are strongly affected by the inhibition of the methyl cycle using the AHCY inhibitor Deazaneplanocin A (DZ), leading to SAH accumulation and inhibition of nucleic acids and protein methylation[27,28]. This link between methylation and circadian rhythms was found in many eukaryotic cells, from unicellular green algae to human cells. These studies concluded that disrupted biological rhythms can be used as a reliable and quantitative sign of methyl metabolism deficiency, with the circadian period increasing proportionally with the concentration of AHCY inhibitors[28].

Here, we used disruptions in circadian rhythms to assess the effects of exogenously administered methyl cycle metabolites, first using a well described in vitro model, then in mouse. These investigations revealed that exogenous SAM is catabolized out of the methyl cycle into adenine, a known toxic metabolite and inhibitor of AHCY, leading to methylation inhibition and circadian rhythms disruption. These results further our understanding of how methyl metabolism works and is regulated, and prompt serious questions about the safety of SAM as a freely available dietary supplement.

## Results

**Exogenous S-adenosylmethionine affects circadian rhythms.** For these experiments, we used mouse embryonic fibroblasts prepared from heterozygous PER2::LUC knock-in mice that express a fusion between luciferase and the endogenous core circadian clock protein Period 2 (PER2)[29]. The circadian clock in these cells can be conveniently measured by real-time luminometry, and the period and robustness of the clock can be extrapolated from the oscillations of PER2::LUC luminescence. As a starting point, we sought to decrease endogenous SAM levels in these cells, speculating that it would affect circadian rhythms in a manner consistent with the pharmacological inhibition of the methyl cycle by AHCY inhibitors like DZ[27,28]. Since SAM is synthesized from the essential amino acid methionine in the methyl cycle (Fig. 1a), methionine deprivation has often been used to trigger methylation deficiencies in vitro[30–32]. We thus tested the effects of methionine concentrations one order of magnitude lower (0.01 mM) or higher (1 mM) than in normal medium.

While no significant effects of 1 mM methionine were seen on the period, 0.01 mM methionine lengthened the period and decreased the amplitude of PER2::LUC oscillations (Fig. 1b). While methionine deficiency is likely to affect many different pathways, notably the mTOR-dependent autophagy pathway, the effect of low methionine observed could at least in part be due to a decrease in SAM. To gain further insights into the interactions between methionine and SAM, the effects of exogenous SAM were investigated in cells cultivated in the presence of various concentrations of methionine. To determine an appropriate range of SAM concentrations, we quantified the intracellular SAM concentration in cells cultivated in standard medium, and obtained a concentration of mean $354 +/-$ S.E.M. 3.9 pmol/$mm^3$ cell volume of SAM (or 0.354 mM), indicating 0.1–1 mM would be an appropriate range to test. Surprisingly, the presence of 1 mM SAM in the medium rescued the cells from low methionine concentration in terms of amplitude and baseline, but the period was consistently lengthened, regardless of the amount of methionine in the medium (Fig. 1c). We confirmed this occurred even in total absence of methionine in the medium (Supplementary Fig. 1a). Only at lower concentrations (0.1 mM), SAM did rescue the cells from low methionine-induced period

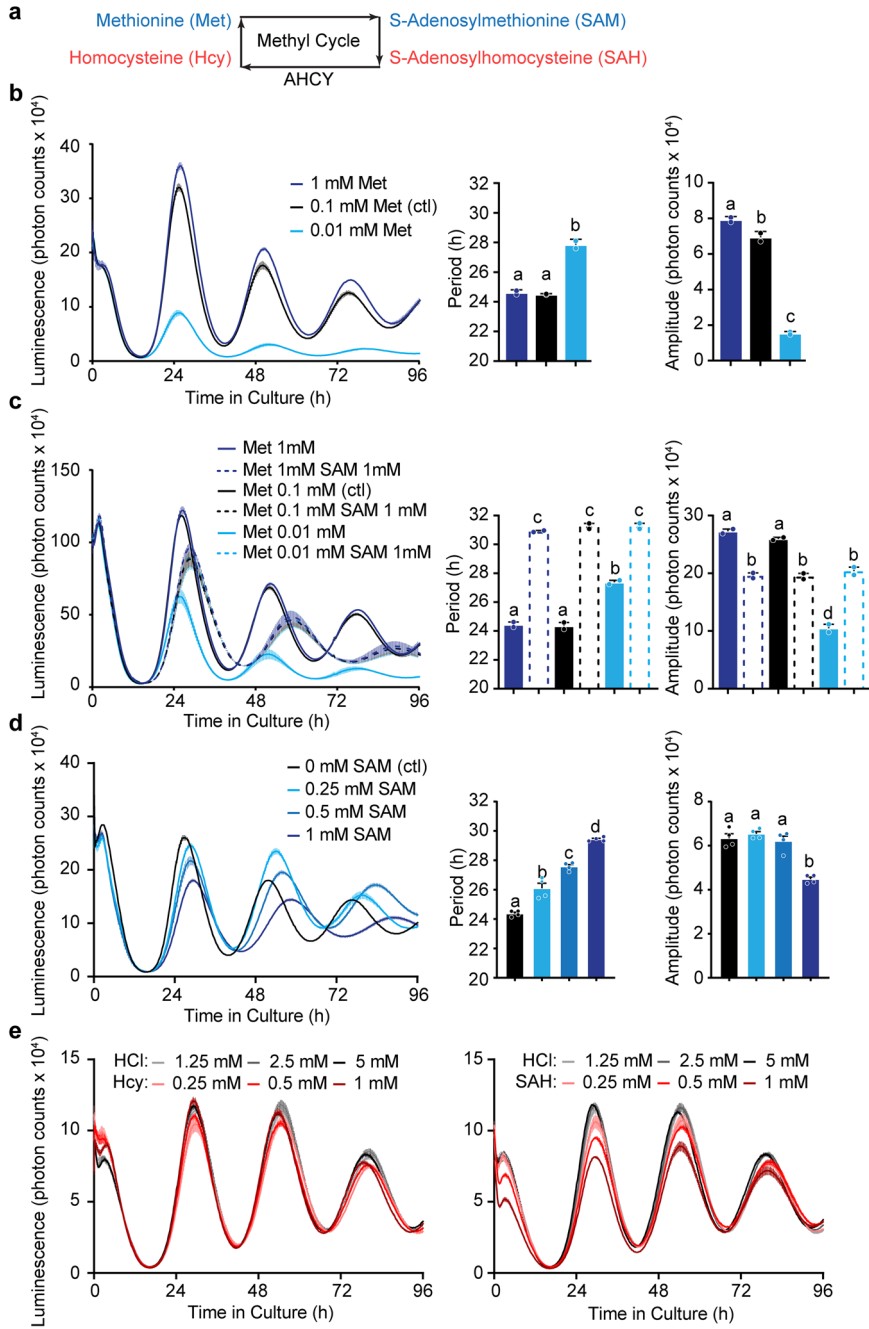

**Fig. 1 Exogenous SAM lengthens the circadian period. a** Simplified representation of the methyl cycle. **b** Low methionine, but not its excess, significantly lengthens the circadian clock period of PER2::LUC oscillations in immortalized mouse embryonic fibroblasts. **c** Partial rescue of the baseline and amplitude drop caused by low methionine with S-adenosylmethionine (SAM), but overriding effects of SAM on period length. **d** Concentration-dependent period lengthening by exogenous SAM. **e** Homocysteine (Hcy) and S-adenosylhomocysteine (SAH) do not lengthen the circadian period compared to the vehicle HCl (Hydrochloric acid). All data shown are mean +/− S.E.M. of 2 (b, 3c) or 4 (d) culture wells. Data shown as bar graphs analyzed by one-way ANOVA followed by Bonferroni multiple comparison test, a vs. b vs. c vs. d (labels at the top of each bar) at least $p < 0.01$.

lengthening (Supplementary Fig. 1b). After 4 days in medium without methionine, cells were obviously under stress and dying, but this was also prevented by SAM (Supplementary Fig. 1c). We next tested SAM at intermediate concentrations (0.25 and 0.5 mM) in an otherwise standard medium: a significant, concentration-dependent period lengthening was observed, up to a maximum of 29.96 +/− S.E.M. 0.2 h at 1 mM SAM (Fig. 1d). In contrast, the other methyl cycle metabolites SAH and homocysteine failed to elicit any significant changes in circadian rhythms (Fig. 1e). SAM affected a human osteosarcoma cell line

(U-2 OS) expressing bioluminescent clock reporter[33] in the same way (Supplementary Fig. 1d), suggesting a conserved mechanism.

The disruption in circadian rhythms caused by exogenous SAM, regardless of the methionine concentration, was an unexpected finding. To understand how cellular metabolism handled exogenous SAM, we compared the intracellular metabolome of cells cultivated in control medium, in the absence of methionine, or in the absence of methionine with 1 mM SAM (Fig. 2a and Supplementary Data 1). As expected, highly significant differences were observed between treatments, with

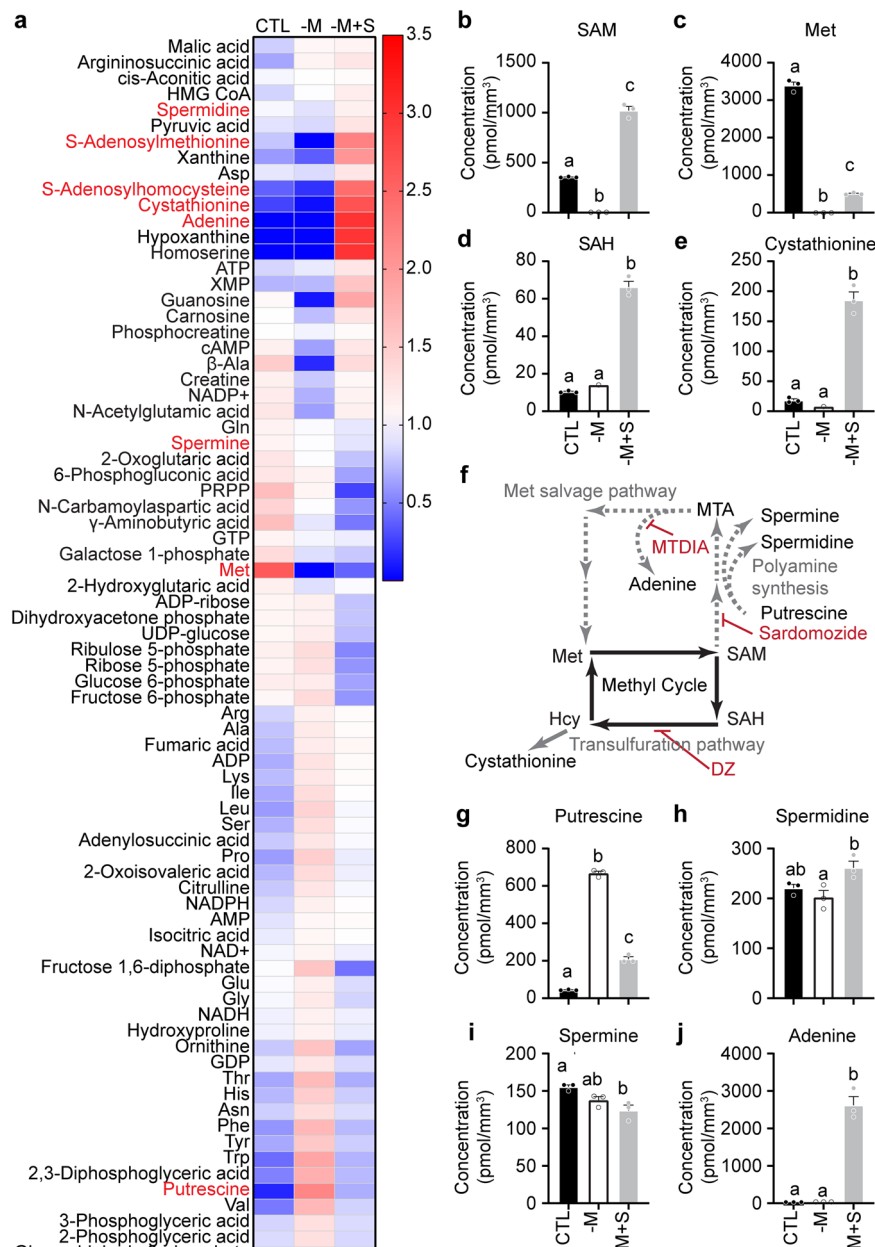

**Fig. 2 Excess SAM is catabolized to SAH and adenine. a** Heatmap representation of metabolites significantly regulated under low methionine without (−M) or with 1 mM SAM (−M + S). Global color scale based on mean +/− S.E.M. of $n = 3$ 10 cm cell culture dishes. Metabolites written in red are shown in the next panels. See also Supplementary Data 1. **b–e** Absolute quantification of SAM, methionine (Met), SAH and cystathionine in cells treated with low methionine or low methionine with 1 mM SAM as in (**a**). **f** Representation of the methyl cycle and its dependents, the methionine salvage and transsulfuration pathways. The methionine salvage pathway also leads to the synthesis of polyamines and to the salvage of adenine from methylthioadenosine (MTA). From cystathionine, the transsulfuration pathway leads to cysteine synthesis. Written in red are the various known pharmacological inhibitors used in this study. **g–j** Absolute quantification of putrescine, spermidine, spermine, and adenine in cells treated with low methionine or low methionine with 1 mM SAM as in (**a**). Data are shown as bar graphs analyzed by one-way ANOVA followed by Bonferroni multiple comparison test, a vs. b vs. c (labels at the top of each bar) at least $p < 0.05$. See also Supplementary Data 1.

free SAM (Fig. 2b) or methionine (Fig. 2c) being undetectable without methionine provided in the medium, and intracellular SAM increasing when provided at 1 mM in the medium. While SAH did not significantly change in methionine-free medium, it significantly increased under 1 mM SAM, which may indicate increased SAM usage (Fig. 2d). This is further supported by increased levels of cystathionine (Fig. 2e), synthesized from homocysteine in the transsulfuration pathway (Fig. 2f) by an enzyme allosterically activated by SAM[34,35]. Consistent with SAM preventing methionine deficiency-induced cell death, in medium

containing 1 mM SAM free methionine was again detected (Fig. 2c), albeit at lower levels than in normal conditions. This occurred most likely because of the methionine salvage pathway, recycling SAM back into methionine when SAM is in excess (Fig. 2f). Indeed, along this pathway we observed significant changes in the polyamines putrescine, spermidine, and spermine (Fig. 2g, h, i). In particular, putrescine dramatically increased in the absence of methionine, indicating the methionine salvage and polyamine synthesis were inhibited under this condition, and partially recovered with the addition of 1 mM SAM.

**Polyamines are likely not involved in period lengthening caused by SAM**. Polyamines homeostasis has been shown to be critical for the circadian clock's function[36]. We thus tested whether an increase in putrescine would lengthen the circadian period in our cells. No effects of putrescine were seen below 1 mM, and only a mild period lengthening was seen at 1 mM (Supplementary Fig. 2a). These data show that increased putrescine levels do not convincingly explain the pronounced period lengthening obtained with SAM.

**Activation of the methionine salvage pathway lengthens the circadian period**. Since exogenous homocysteine (Fig. 1e), SAH (Fig. 1e), methionine (Fig. 1b), or putrescine (Supplementary Fig. 2a) were not associated with period lengthening, we hypothesized that either SAM itself, or another metabolite along the methionine salvage pathway, was responsible for the period lengthening observed. Along the methionine salvage and after the polyamine branch, the next metabolite is 5′-methylthioadenosine (Fig. 2f, MTA), a sulfur-containing purine nucleoside present in all mammalian tissues that has been used pharmacologically to regulate gene expression and cell proliferation, differentiation, and apoptosis[37]. Increased levels of MTA when the methionine salvage is activated could contribute to the effects of SAM on circadian rhythms. While MTA was not measured in Fig. 2, adenine, a product of MTA phosphorylation in the methionine salvage pathway, dramatically increased in the presence of SAM (Fig. 2j), suggesting MTA did also increase. In fact, this is the only known metabolic route from MTA in mammals: MTA phosphorylation by MTA-phosphorylase (MTAP) to yield 5-methylthioribose-1-phosphate and adenine. MTAP is a clinically relevant enzyme in the treatment of cancer since the reaction it mediates is also key for purine nucleotide salvage that is overactivated in cancer cells[38–40]. This also explains why, in cells cultivated in the presence of 1 mM SAM, the purine bases xanthine, adenine, hypoxanthine, as well as the purine nucleoside guanosine and nucleotide XMP, all products of purine salvage, significantly increased (Fig. 2a). Activation of the purine salvage is known to block de novo purine synthesis, since adenine allosterically inhibits the rate-limiting step to 5-phosphoribosyl 1-pyrophosphate (PRPP), as well as that of its own branch to AMP[41] (Supplementary Fig. 2b). Consistent with this, PRPP significantly decreased under 1 mM SAM (Fig. 2a), accompanied with an increase in XMP (Fig. 2a), the first nucleotide in the alternative branch of de novo purine synthesis (Supplementary Fig. 2b).

Since 1 mM methionine in the culture medium did not affect the period (Fig. 1b), the lengthening of the period seen with SAM is probably not dependent on metabolites of the methionine salvage further downstream of MTA and adenine. Rather, the activation of adenine salvage seemed a good candidate for further investigations. To test this, we treated PER2::LUC MEFs with 0.25, 0.5, and 1 mM of MTA or adenine, and observed concentration-dependent period lengthening of comparable magnitude with that observed with SAM (Supplementary Fig. 3a, b). This clearly demonstrates the importance of this pathway for circadian rhythms.

To gain further insights on the intracellular metabolic response to exogenous SAM and adenine, and the mechanisms underlying period lengthening, we compared the intracellular metabolome of cells treated with 1 mM SAM or adenine (Fig. 3a and Supplementary Data 2). This time, a wider set of metabolites was analyzed semi-quantitatively. The effects of 1 mM SAM on the methyl cycle and methionine salvage metabolites were consistent with the previous data, with an increase in methionine (Fig. 3b), SAM (Fig. 3c), SAH (Fig. 3d), and cystathionine (Fig. 3e). In contrast, 1 mM adenine had no

significant effects on methionine, cystathionine, or polyamines, but also increased levels of SAM and SAH. This time we confirmed an increase in MTA under 1 mM SAM treatment (Fig. 3f), as well as in adenine (Fig. 3g), further evidence that the methionine salvage is activated by SAM. Changes in purine nucleosides and bases elicited by 1 mM SAM or adenine were consistent with previous observations, and both 1 mM SAM and adenine increased intracellular levels of adenine, hypoxanthine, and guanosine, together with all purine nucleotides, despite near undetectable levels of PRPP and AICAR (Fig. 3a), two key metabolites of the de novo purine synthesis pathway (Supplementary Fig. 2b). These data demonstrate that the salvage of purines has been activated similarly by SAM and adenine. Since PRPP is also required for the synthesis of pyrimidine nucleotides (Supplementary Fig. 2c), virtually all pyrimidines detected (UDP, UTP, UMP, CTP, CDP, dTTP…) decreased under 1 mM of either SAM or adenine (Fig. 3a).

**Purine/pyrimidine imbalance does not explain period lengthening**. An inhibitory effect of excess adenine on the cell cycle has previously been proposed to be mediated by a purine/pyrimidine imbalance, an inhibition that could in some cases be rescued by adding the pyrimidine nucleoside cytidine[42]. Reasoning that, since the cell cycle is itself controlled by the circadian clock[43], the lengthened circadian period could also be rescued by exogenous cytidine, we treated cells with adenine together with equimolar amounts of cytidine or its base cytosine, but observed no effects of the pyrimidines, alone or together with the purine bases (Fig. 3h). We concluded that purine/pyrimidine imbalance does not readily explain the period lengthening. This is similar to other reports about the toxicity of adenine shown to not depend on the purine/pyrimidine imbalance[44], or on any nucleotide, for example ATP or GTP, that could be salvaged from adenine[45]. Since these reports also showed that excess adenosine, like adenine, was toxic to cells, we tested whether adenosine would also lengthen the circadian period but did not observe any significant effects (Fig. 3i).

**Adenine is an endogenous AHCY inhibitor**. So far we have demonstrated that purine-based metabolites derived from SAM have the ability to affect circadian rhythms, but the underlying mechanisms remain unknown. In our metabolome analyses, we noticed that, although SAM increased when 1 mM SAM or adenine were provided in the medium (and in the absence of methionine with 1 mM SAM), SAH increased at least one order of magnitude higher than SAM, causing the [SAM]/[SAH] ratio to fall (Fig. 4a). [SAM]/[SAH], often called the "methylation potential," is an indicator of methylation status, and should remain high to ensure methylations can occur[46]. This suggests excess of these metabolites may negatively feedback on the methyl cycle, inhibiting SAH hydrolysis to homocysteine by AHCY, leading to SAH accumulation, protein and mRNA methylation inhibition and period lengthening as observed with bona fide inhibitors of AHCY such as DZ (itself a structural analog of adenine nucleoside)[27,28]. This hypothesis is supported by biochemical literature published in the late 70s reporting that adenine and MTA may bind to and inhibit AHCY[44,47,48]. To test whether MTA, adenine, and SAM do regulate AHCY activity, we performed in vitro enzymatic assays with purified AHCY. Only adenine, like DZ used as a positive control, showed a clear and concentration-dependent AHCY inhibition (Fig. 4b). MTA mildly but significantly inhibited AHCY, albeit no significant dose-dependent effects were seen at the concentrations tested (Fig. 4b). These data support the hypothesis that activation of the methionine salvage pathway by exogenous SAM increases levels

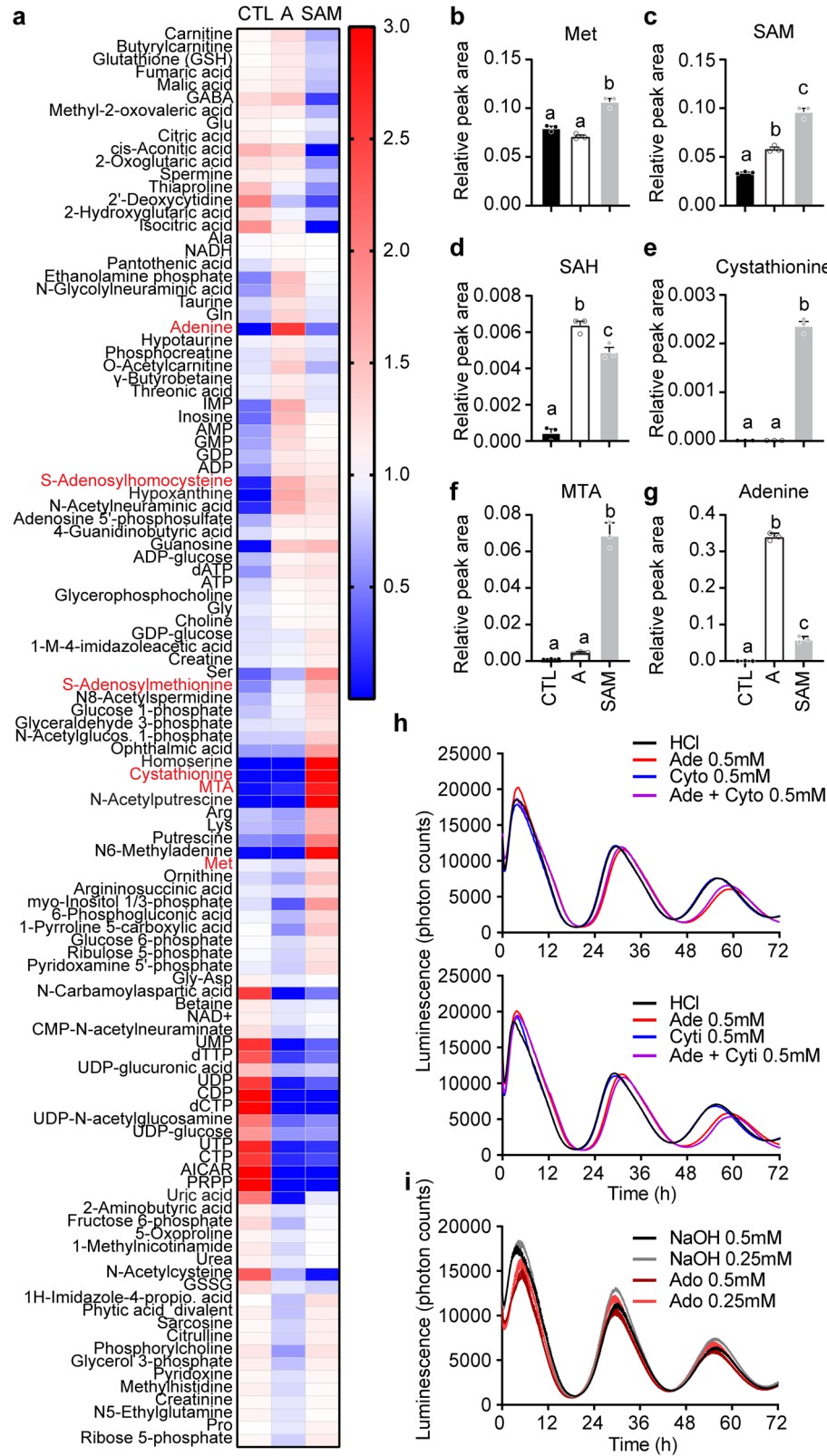

of MTA and adenine, inhibiting AHCY and leading to circadian period lengthening.

To confirm methyl metabolism reacts the same way in cells treated with SAM, MTA, DZ, and adenine, we measured the levels of adenine, SAM, SAH, and MTA in cells treated with these molecules by LC/MS-MS (Fig. 4c). SAM and MTA, as predicted, led to the accumulation of adenine, undetectable under control

conditions. Consistent with our predictions, SAH dramatically increased under DZ, SAM, adenine, or MTA treatment.

**AHCY overexpression changes the sensitivity of cells to adenine, SAM, and MTA.** If AHCY inhibition is the major mechanism underlying period lengthening obtained with adenine, MTA, or SAM, then increasing AHCY expression should

**Fig. 3 Adenine and SAM disrupt the purine/pyrimidine ratio, but complementation by pyrimidines does not rescue the cells from period lengthening.**
**a** Heatmap representation of metabolites significantly regulated under 1 mM adenine (A) or 1 mM SAM (SAM). Red and blue represents higher and lower values, respectively (see legend on the right). Metabolites written in red are shown in the next panels. Note the parallel changes in SAH, purine, and pyrimidine nucleotides in adenine and SAM-treated cells. Global color scale based on mean+/−S.E.M. of $n = 3$ 10 cm cell culture dishes. See also Supplementary Data 2. **b**–**g** Relative quantification of intracellular Met, SAM, SAH, cystathionine, MTA and adenine in cells treated with 1 mM adenine or 1 mM SAM. Data show mean +/− S.E.M. of $n = 3$ cell culture wells. **h** Cytosine (Cyto) or cytidine (Cyti) do not rescue PER2::LUC mouse embryonic fibroblasts from the period lengthening effects of adenine (Ade). Data show mean +/− S.E.M. of $n = 4$ cell culture wells. **i** In contrast to adenine, adenosine (Ado) does not significantly lengthens the circadian period. Data show mean +/− S.E.M. of $n = 4$ cell culture wells. All data shown as bar graphs analyzed by one-way ANOVA followed by Bonferroni multiple comparison test, a vs. b vs. c (labels at the top of each bar) at least $p < 0.05$. See also Supplementary Data 2.

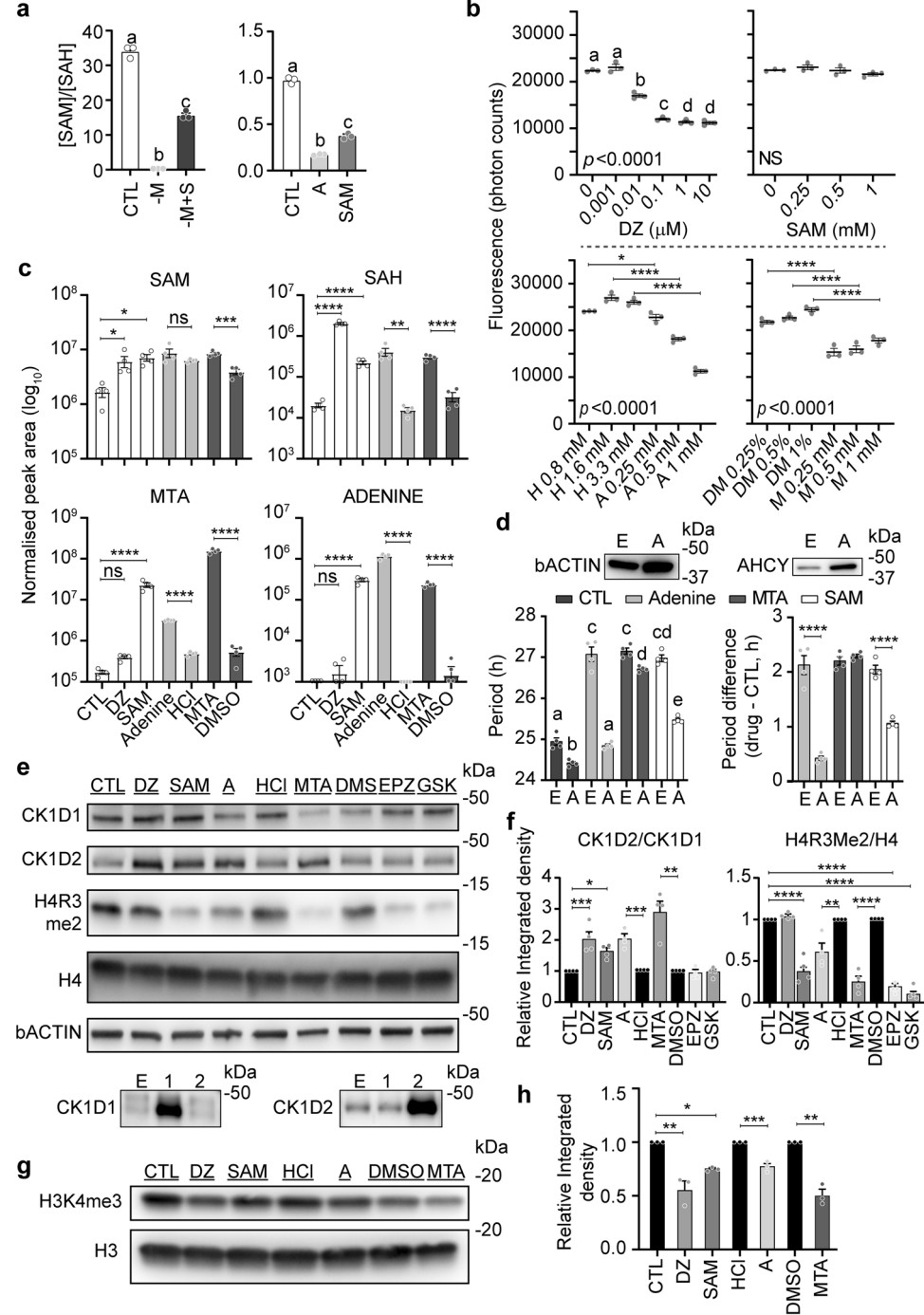

**Fig. 4 Exogenous SAM, via its catabolism to MTA and adenine, inhibits methyl metabolism. a** SAM and adenine similarly decrease the SAM/SAH ratio. Data show mean $+/-$ S.E.M. of $n = 3$ 10 cm cell culture dishes. The differences in the Y axis values originate from the type of raw data from these two metabolome datasets, i.e., absolute quantification (left) versus semi-quantitative (right). **b** In vitro enzymatic assays demonstrate that adenine and MTA are direct AHCY inhibitors. DZ was used as a positive AHCY inhibitor control, and SAM did not have any significant effects. Data show mean $+/-$ S.E.M. of $n = 4$ assay wells. HCl (H) was used as a vehicle for adenine (A), DMSO (D) for MTA (M), and water for DZ and SAM. DZ and SAM treatments analyzed by one-way ANOVA followed by Bonferroni multiple comparison test, a vs. b vs. c vs. d (labels at the top of each bar) $p < 0.0001$; for adenine and MTA, only the indicated comparisons were made. **c** Quantification of SAM, SAH, MTA, and adenine by LC-MS/MS reveals adenine accumulates in cells treated with SAM, adenine, or MTA (all at 1 mM), leading to an increase in SAH as observed for DZ treatment (10 μM). Data show mean peak area normalized to total cell volume $+/-$ S.E.M. of $n = 4$ 10 cm cell culture dishes, analyzed by one-way ANOVA followed by Bonferroni multiple comparison test (CTL vs. DZ vs. SAM) or unpaired $t$ test for adenine vs. HCl and MTA vs. DMSO, due to the effects of HCl and DMSO on cell proliferation. Although the increase of SAH in cells treated with SAM was not significant in Bonferroni multiple comparison test after ANOVA due to the much stronger effect of DZ, in $t$ test CTL vs. SAM was highly significant, with $p < 0.0001$. CTL was vehicle (water) for DZ and SAM. HCl and DMSO were vehicle controls for adenine and MTA, respectively. **d** Overexpression of AHCY protects the cells against adenine and SAM. Left graph shows mean period $+/-$ S.E.M. of $n = 4$ replicate dishes compared by one-way ANOVA followed by Bonferroni multiple comparison test, a vs. b vs. c vs. d vs. e (labels at the top of each bar) at least $p < 0.05$. Right graph shows mean drug − CTL period difference $+/-$ S.E.M. of $n = 4$ replicate dishes compared by one-way ANOVA followed by Bonferroni multiple comparison test between each EMPTY vector (E) and AHCY (A) couple. Expression of AHCY was confirmed by immunoblotting, shown above the graphs. **e** Immunoblotting reveals that SAM, adenine (A), and MTA (all at 1 mM) stimulate CK1D2 expression, a hallmark of AHCY inhibition by DZ confirmed here (10 μM). In contrast, specific inhibition of H4R3 symmetric demethylation (H4R3me2) by PRMT5 with EPZ (10 μM) or GSK (10 μM) does not affect CK1D1 or CK1D2 expression. H4 and bActin were used as loading controls, and the specificity of CK1D1 and CK1D2 antibodies was ascertained using extracts from cells transfected with their respective expression vectors, results shown at the bottom. **f** Bar charts show mean CK1D2/CK1D1 or H4R3me2/H4 ratio of relative integrated densities $+/-$ S.E.M. of $n = 4$ independent membranes; CTL, DZ, SAM, EPZ, and GSK analyzed by one-way ANOVA followed by Bonferroni multiple comparison test (ratios for CTL was expressed as 1), and A vs. HCl (ratios for HCl expressed as 1) and MTA vs. DMSO (ratios for DMSO expressed as 1) separately analyzed by $t$ test. **g** Immunoblotting showing DZ (10 μM), SAM, adenine, and MTA (all at 1 mM) decrease H3K4me3. **h** Bar chart shows mean H3K4me3/H3 ratio analyzed as in (**f**), $n = 3$ independent membranes. Significances throughout are *$p < 0.05$, **$p < 0.01$, ***$p < 0.001$, ****$p < 0.0001$. See Supplementary Fig. 7 for uncropped blots.

make the cells less sensitive to these metabolites (i.e., raise the IC50). To test this possibility, we treated PER2::LUC cells stably transfected with an expression vector for AHCY with a low concentration of adenine, MTA, or SAM (250 μM). Interestingly, AHCY overexpression alone shortened the circadian period, and cells overexpressing AHCY were less sensitive to adenine and SAM compared with cells stably transfected with the empty vector (Fig. 4d), demonstrating AHCY inhibition is one of the main factors underlying the effects of these metabolites on circadian rhythms. Overexpression of AHCY did not protect the cells against MTA, however, suggesting alternative AHCY-independent mechanisms underlying period lengthening when cells are treated with MTA.

**Inhibition of the methionine salvage pathway rescues the cells from SAM**. Next, to prevent adenine and/or MTA synthesis from SAM, we treated PER2::LUC cells with sardomozide, an inhibitor of SAM decarboxylase[49] (Fig. 2f), the first and rate-limiting enzyme in the methionine salvage pathway[50], or MTDIA (Methylthio-DADMe-Immucillin A), an inhibitor of MTAP[51,52] (Fig. 2f), together with different concentrations of SAM. In a manner consistent with our results so far, sardomozide and to a lesser extend MTDIA protected the cells from the period lengthening effects of SAM (Supplementary Fig. 4a, b). Without excess SAM, sardomozide shortened the period compared to untreated control (Supplementary Fig. 4a), indicating that the normal rate of the methionine salvage pathway is important for clock function, while MTDIA weakly lengthened the period (Supplementary Fig. 4b) probably due to the resulting MTA accumulation[51]. Together these data demonstrate that the period lengthening caused by SAM depends on its catabolism to MTA and adenine.

**MTA directly inhibits PRMT5**. While we have shown that adenine and to a lesser extent MTA directly inhibit AHCY, leading to SAH accumulation associated with circadian period lengthening, more direct mechanisms may contribute to the

period lengthening effects of these metabolites. In particular, MTA has been reported to directly inhibit PRMT5-mediated symmetric dimethylation of H4R3 (H4R3Me2)[53], which could potentially also affect the circadian clock in mammalian cells as it does in plants[54]. In vitro, PRMT5 assay confirmed that only MTA had a consistent inhibitory effect on the symmetric dimethylation of a H4R3 peptide, while SAM, as the methyl donor co-substrate, promoted H4R3 methylation in vitro (Supplementary Fig. 5a). In the cell, however, high SAM levels will lead to MTA accumulation, likely inhibiting PRMT5 as well (see below). To test the contribution of PRMT5 inhibition to period lengthening, we tested the two specific PRMT5 inhibitors GSK591 and EPZ015666 in PER2::LUC MEF cultures. Both inhibitors have an IC50 of ~10 nM for the methylation of histone 4 (H4), but only a mild effect on the period was observed for GSK591 at 10 μM (Supplementary Fig. 5b), indicating PRMT5 may not be a strong regulator of circadian rhythms in this system. Considering the lack of rescue afforded by AHCY overexpression (Fig. 4d), MTA may directly affect other MTases, targeting arginine or other substrates, relevant for circadian rhythms. Indeed, MTA has been shown to widely inhibit arginine methylation[55].

**Hallmarks of AHCY inhibition: CK1D expression changes, decrease in histone methylation**. A hallmark of the cellular response to DZ is a shift in the relative expression of two alternatively spliced isoforms of casein kinase 1 delta (CK1D)[56]. Therefore, if SAM, adenine, and MTA lead to widespread methylation inhibition like DZ, they should also trigger similar changes in CK1D expression. We proceeded to test this possibility by immunoblotting and, consistent with our hypothesis, after a 24h-treatment of PER2::LUC MEFs, SAM, MTA, and adenine, like DZ, increased the CK1D2/CK1D1 expression ratio (Fig. 4e, f). This increased CK1D2/CK1D1 ratio in itself could explain the period lengthening observed with adenine and other AHCY inhibitors, as previously shown[56].

To confirm SAM, MTA and adenine lead to methylation inhibition, we investigated their effects on histone methylation. The first histone methyl mark we measured was H4R2me2, since we

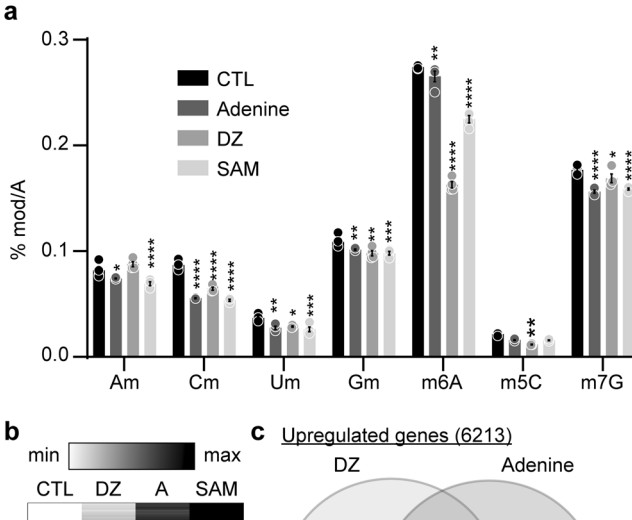

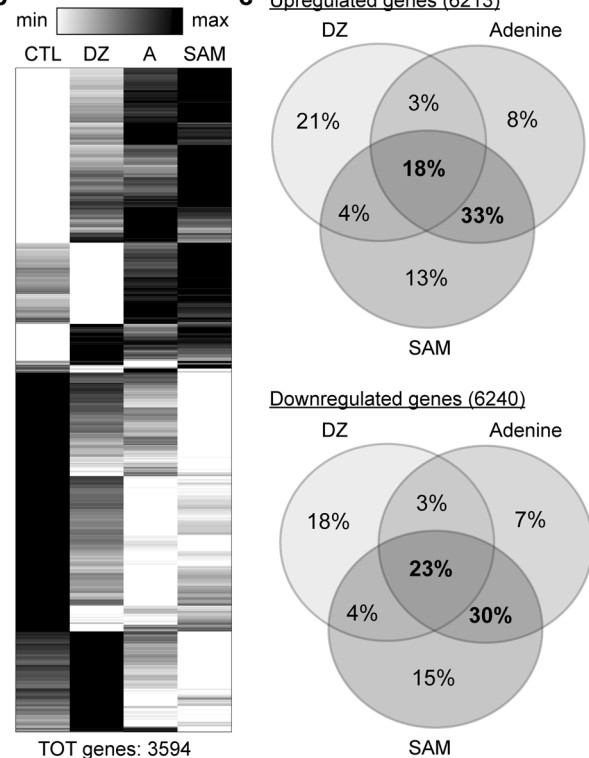

**Fig. 5 Adenine and SAM inhibit mRNA methylation and have similar transcriptome signature. a** Quantification of methylated nucleotides in mRNA from cells treated with DZ, adenine, or SAM. Am, Cm, Um, and Gm correspond to 2′-O-ribose methylation of A, C, U, and G, respectively; m6A is $N^6$-methyladenosine; m5C is 5-methylcytosine; m7G is $N^7$-methylguanine. Data show mean +/− S.E.M. of $n = 4$ cell culture wells, analyzed by two-way ANOVA (all sources of variations $p < 0.0001$) followed by Bonferroni multiple comparison test, $^*p < 0.05$, $^{**}p < 0.01$, $^{***}p < 0.001$, $^{****}p < 0.0001$. **b** Heatmap representation of 3594 genes significantly regulated by DZ (10 μM), adenine (1 mM), and SAM (1 mM), based on the mean of 3 replicate dishes. **c** Venn diagrams showing the overlap in significantly up- (6213) and downregulated (6240) genes between cells treated with DZ (10 μM), adenine (1 mM), or SAM (1 mM). See also Supplementary Data 3.

showed earlier that PRMT5 is directly inhibited by MTA in vitro (Supplementary Fig. 5a), and confirmed that a 24 h-treatment of PER2::LUC MEFs with SAM, MTA, or adenine caused a decrease in H4R2me2, while DZ had no effects (Fig. 4e, f). This is in line with the inhibition of PRMT5 being caused mainly by MTA, not by SAH. GSK591 and EPZ015666 were used as positive controls.

We next tested H3K4Me3, a histone mark that has been shown to contribute to the AHCY-dependent control of cyclic gene expression by the circadian clock and to be inhibited by DZ[57–59]. As expected, H3K4Me3 decreased in cells treated for 24 h with DZ, adenine, MTA, or SAM (Fig. 4g, h).

These data further underline the similarities between the effects of DZ, adenine, MTA, and SAM. In addition to "DZ-like" AHCY inhibition caused by adenine, the catabolism of excess SAM also leads to MTA accumulation that increases the range of inhibited methylation reactions. While MTA is relevant for the toxicity of excess SAM, we omitted MTA from our next experiments, since MTA seems to have more AHCY-independent modes of action compared to DZ and adenine.

**Hallmarks of AHCY inhibition: widespread inhibition of mRNA methylation**. We have previously reported that inhibition of AHCY with DZ or analogs leads to the inhibition of $N^6$-methyladenosine in mRNA[27,28,56]. To confirm whether this also occurred in cells treated with adenine and SAM, we extracted mRNA from PER2::LUC cells treated with these metabolites for 24 h, using DZ as a positive control, and quantified all major methylated nucleotides by LC-MS (Fig. 5a). A significant and widespread decrease in methylated nucleotides was observed with adenine, SAM, and DZ, demonstrating they ultimately lead to methylation inhibition.

**DZ, adenine, and SAM have similar transcriptome signatures**. Despite differences between DZ, adenine, and SAM in the pathways they potentially regulate in the cell, the fact that they converge in the inhibition of AHCY suggest they should elicit similar transcriptome signatures in cells treated with these molecules. Since, as we have shown above, excess SAM is converted to adenine in the cell, these two metabolites should have uncanny similarities in the genes they regulate. We thus proceeded with analyzing transcriptomes by RNASeq. As expected DZ, adenine, and SAM had profound effects, with 5825, 7795, and 8767 genes significantly regulated (false discovery rate FDR < 0.01) among a total of 12,591 expressed genes, respectively (Supplementary Data 3). A heatmap containing the 3594 genes that significantly vary in all treatments shows the similarity in transcriptome signatures between DZ, adenine, and SAM (Fig. 5b), and Venn diagrams with genes significantly up- and downregulated in at least one treatment further show that ~20% of genes are shared among the three treatments, and an additional 30% of genes are similarly regulated by adenine and SAM (Fig. 5c). A stringent gene ontology (GO) analysis based on significantly regulated genes (FDR < 0.01) in each treatment reveals a profound overlap between affected biological processes (Supplementary Data 4). Consistent with the common effects of DZ, adenine, and SAM on methyl metabolism and circadian rhythms, "Methylation" (GO:0032259) and "Rhythmic process" (GO:0048511) were highly significant ontologies in each treatment, and "Circadian rhythms" (GO:0007623) was also significant in both adenine- and SAM-treated samples. Together these data show that exogenous SAM, despite being thought of as beneficial, elicit changes in gene expression that share high similarity with the response to known toxic metabolites and methyl metabolism inhibitors like adenine or DZ.

**SAM is highly toxic to dividing cells**. The circadian experiments above were performed with confluent cell populations of PER2::LUC MEFs, which are contact-inhibited. This allows for stable and robust luminescence rhythms to be measured. In these populations, no effects of SAM on the cell cycle or cell death were seen. The anti-proliferative and pro-apoptotic effects of SAM in cancer cells[24–26],

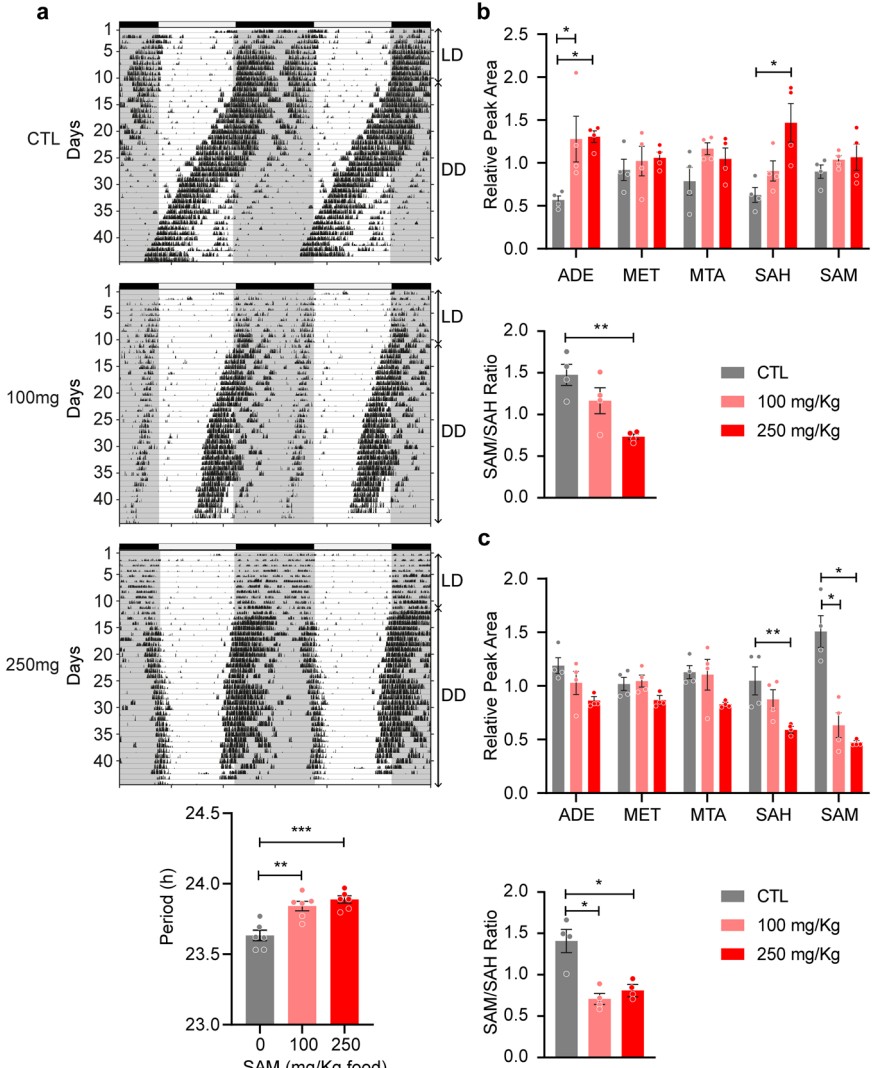

**Fig. 6 SAM supplementation affects circadian behavior in vivo. a** Representative running wheel activity records (actograms) of mice fed control (CTL) diet, or the same diet enriched with 100 or 250 mg/kg food, as indicated. Actograms are double-plotted, first recorded under standard light-dark conditions (LD, days 0 to 10) followed by constant darkness (DD, days 11 to 45). Black tick marks inside the actograms represent bouts of wheel-running activity. Periods of light and dark are indicated by horizontal bars above actograms, and the timing of the original light-dark cycles is indicated by grayed areas. Bar chart shows mean circadian period +/− S.E.M. of $n = 6$ male mice, analyzed by one-way ANOVA ($p < 0.001$) followed by Bonferroni multiple comparison test, \*\*$p < 0.01$, \*\*\*$p < 0.001$. See Supplementary Fig. 8 for raw actograms of all animals. **b** Metabolites quantification from the liver and hypothalamus (**c**) of mice fed control diet, or the same diet enriched with SAM at 100 or 250 mg/kg food for 3 weeks. Lower graphs in each panel show SAM/SAH ratio. Bar chart shows mean +/− S.E.M. of $n = 2$ male mice + 2 female mice, analyzed for each metabolite by one-way ANOVA followed by Bonferroni multiple comparison test, \*$p < 0.05$, \*\*$p < 0.01$, significance only shown when ANOVA and multiple comparison tests are $p < 0.05$.

however, prompted us to test whether SAM affected the growth of PER2::LUC MEFs. Cells cultivated at low confluence were treated with 1 mM SAM or vehicle, and counted every day for 5 days. While control cells grew exponentially, cells treated with SAM initially showed stunted growth followed by a rapid decline (Supplementary Fig. 6a). These results further highlight SAM as a toxic metabolite with potential uses as an anti-neoplastic agent.

**SAM supplementation affects circadian behavior in vivo**. The metabolism of SAM in vivo is more complex than in cell culture, and the methionine salvage pathway may not be active in all tissues. Enteric SAM may thus not have the same detrimental effects as in vitro. To match the daily recommended intake of SAM supplements of 0.8–1.6 g/day in human, we administered SAM to mice at a concentration of 100 and 250 mg/kg of food,

and monitored circadian locomotor (running wheel) activity rhythms in constant darkness for around 1 month to allow expression of endogenous rhythms. At both 100 and 250 mg/kg, SAM caused an increase in the circadian period compared to control, with a dose-dependent effect seen by an increase in the CTL vs. SAM significance in post hoc analysis (Fig. 6). While mice did not display any obvious differences in general condition including body weight after one month under these conditions, the fact that circadian rhythms were affected by exogenous SAM as observed in vitro indicate unsupervised SAM supplementation should be considered with caution.

To gain insights into how SAM intake affects systemic methyl metabolism, we quantified selected metabolites from the liver and hypothalamus of mice fed for three weeks with SAM at 100 and 250 mg/kg and compared with animals fed with unsupplemented control diet (Fig. 6b, c). As expected from in vitro experiments,

adenine increased in the liver, indicating the methionine salvage pathway is active in this tissue. Interestingly, the SAM/SAH ratio decreased in both tissues, but for different reasons: in the liver, SAH increased but SAM did not change, while in the brain both SAM and SAH decreased so that SAH > SAM. This clearly shows that SAM supplementation negatively disrupts systemic methyl metabolism.

In conclusion, using disrupted molecular and behavioral circadian rhythms as a read-out for methyl metabolism deficiency, we show that the methyl cycle is highly sensitive to exogenous variations of its metabolites, and that excess SAM beyond endogenous levels, rather than promoting methylation, has negative consequences due to its catabolism to MTA and adenine by the methionine salvage pathway. We show these SAM catabolites are potent AHCY and MTases inhibitors. This calls for a tightening of legislation regarding the availability of SAM as an over-the-counter dietary supplement.

## Discussion
Beyond mRNA and histone methylation, there may be other MTases directly or indirectly inhibited by excess of SAM, adenine, and MTA. These may include MTases targeting DNA, tRNA, and rRNA. A comprehensive analysis of all methylated residues in proteins and nucleic acids potentially affected is out of the scope of the present work. The main conclusion here, that excess SAM is catabolized to MTA and adenine, leading to methylation deficiencies, stands.

While we have flagged the potential toxicity of SAM, the effects of SAM intake on locomotor activity rhythms in vivo suggest SAM could potentially be used to correct circadian misalignments caused by shiftwork or jetlag. Circadian rhythm sleep disorders such as delayed sleep phase disorder (DSPD) and advanced sleep phase disorder (ASPD) are often associated with depression[60,61]. SAM may be especially efficient in the treatment of ASPD, since ASPD patients have a circadian clock that runs faster than 24 h, leading to early sleep onset and awakening. SAM may help slow the internal rhythms of ASPD patients down and facilitate their synchronization to the external day-night cycles. This also suggests that future clinical trials using SAM to treat depression may need to consider chronotypes as a confounder that may have been the origin of failed clinical trials.

Given the existing literature on adenine/adenosine toxicity, we were surprised that the adenosine nucleoside did not affect the clock like its base (Fig. 3i): Together with homocysteine, adenosine is the product of SAH hydrolysis by AHCY, and thus a substrate for the reverse reaction when other routes of adenosine metabolism are interrupted, causing an increase in SAH[62,63]. The clinical relevance of adenine and adenosine toxicity in the etiology of *Adenosine deaminase* deficient-severe combined immunodeficiency (ADA-SCID)[64,65] provided us with a clue. Under normal conditions, the enzyme ADA efficiently deaminates adenosine to inosine, an important step in the catabolism of adenosine. Highest levels of *Ada* are found in cells of the lymphoid system, explaining why ADA deficiency causes SCID, and why ADA is a common target for leukemia chemotherapies. Many investigations into the mechanisms of adenosine toxicity in vitro are based on human lymphoblasts treated with ADA inhibitors in conjunction with exogenous adenosine. We thus tested whether adenosine, together with the specific *Ada* inhibitor pentostatin (Nipent™, Pfizer) currently used in the treatment of various types of leukemia[66], would elicit period lengthening. While no significant effects of adenosine or pentostatin alone were observed on the circadian period, a dramatic period lengthening, which was dependent on the concentration of adenosine, was observed

when the two compounds were used together (Supplementary Fig. 6b), strengthening our conclusions and suggesting methylation deficiency affecting the circadian clock could contribute to the disrupted sleep of ADA-SCID patients[67,68].

Oral adenine is classified as acutely toxic and is often used to induce chronic kidney disease in rodents, caused by the catabolism of adenine into 2,8-dihydroxyadenine that crystallizes in renal tubules[69]. Excess adenine intake to induce nephropathy in mouse was shown to affect circadian behavior in vivo, but no mechanisms were proposed[70,71]. Our results suggest that adenine per se may affect circadian rhythms in these mice via a methylation-dependent manner. More investigations should confirm this possibility.

## Methods
**Animals**. Animal experiments were licensed under the Animals (Scientific Procedures) Act of 1986 (UK) and were approved by the animal welfare committees at the University of Manchester. Eight- to twelve-week-old C57BL/6J male mice were purchased from Charles River (UK) and acclimatized to the local animal unit for 1 week before the experiment. Mice were housed under a 12:12 h light/dark cycle at ~350 lux during the light phase and 0 lux during the dark phase. Ambient temperature was kept at 22 ± 3 °C, the humidity was ~52 ± 4%, with food and water available ad libitum. Meal diets containing 127 mg/kg or 318 mg/kg SAM chloride dihydrochloride (Sigma A7007) were ordered from Special Diets Services, using their standard maintenance diet #BK001 as a base. Control diet was the same meal diet, BK001. Running wheel activity rhythms were measured in single-mouse cages equipped with running wheels (Vet Tech Solutions Ltd) placed in light-tight programmable cabinets (Tecniplast). Actograms and period estimates (chi-square periodogram) were acquired with Clocklab (Actimetrics).

**Cell cultures**. Mouse PER2::LUC[29] cells or Bmal1-luc U-2 OS[33] were cultivated as previously described[27] with the following alternative modifications. Cells were seeded on 35 mm dishes (Corning) or 24-well plates (Corning) and allow to grow to confluence in DMEM/F12 medium (Invitrogen 31330038) containing antimycotic/antibiotic (Sigma A5955) and heat-inactivated serum (Gibco 10500064). Cells were then shocked by dexamethasone (Sigma-Aldrich D4902) 400 nM for 2 h, followed by a medium change including 1 mM beetle luciferin (Promega E1605). 35 mm dishes were then transferred to a luminometer (Lumicycle32, Actimetrics) and 24-well plates to a luminometer/incubator (CL24A-LIC, Churitsu Electric Corp.). Photons were counted in bins of 2 min at a frequency of 10 min. Period and amplitude were estimated by BioDare2[72]. DZ (SML0305), adenine (A2786), 1 M HCl (H9892), MTA (260585), DMSO (D8418), SAM (A7007), GSK591 (SML1751) and EPZ015666 (SML1421) were purchased from Sigma-Aldrich; Methylthio-DADMe-Immucillin A (HY-101496) and Sardomozide (HY-13746B) were purchased from Cambridge Bioscience Ltd. Stocks for DZ (10 mM in water), adenine (150 mM in 0.5 N HCl), HCl (0.5 N), MTA (100 mM in 100% DMSO), DMSO (100%), SAM (100 mM in water), GSK591 (10 mM in 100% DMSO), EPZ015666 (10 mM in 100% DMSO), MTDIA (10 mM in 100% DMSO), and Sardomozide (10 mM in water) were kept in −30 °C.

For the cell growth experiment, PER2::LUC cells were plated at 5000 cells/well in twelve 6-well plates, 4 wells per plate. The next day, medium was replaced with medium containing 1 mM SAM or vehicle, 6 plates for each treatment. Two plates, one for control one for SAM, were immediately washed once with 2 ml PBS, trypsinised (300 μl trypsin/well, Sigma T3924), and incubated 3 min in the incubator. Cells were transferred to sterile 1.5 ml tubes and immediately counted using a manual hemocytometer. This was repeated every day for 5 days with the remaining sets of plates.

**DNA vectors**. Expression vector for AHCY was obtained by amplifying the mouse AHCY ORF from the HA-AHCY vector[59] using the primers 5′-ATAGTC-GACGCCACCATGTACCCATACGATGTT-3′ and 5′-TATGCTAGCTTCAG-TAGCGGTAGTGATCAGGCT-3′, then ligating the SalI/NheI-digested PCR product into the SalI/NheI-digested pSELECT-HYGRO-MCS vector (Invivogen) using Ligation high Ver.2 (Toyobo).

**Immunoblotting**. Confluent cells cultivated in 24-well plates were treated with each respective chemical for 24 h in the incubator at 37 °C, 5% CO₂. Cells were washed once with 1 ml PBS then lysed in the plate with 0.1 ml/well 2X Laemmli buffer (Bio-Rad) supplemented with 20 mM DTT. Cells were scraped out with a pipette tip and transferred to a 1.5 ml microtube, boiled for 10 min at 95 °C, vortexed at full speed for 5–10 s, and spinned-down before split into single-use aliquots kept at −30 °C. On the day of the immunoblotting, aliquots were boiled again for 10 min at 95 °C, vortexed at full speed for 5–10 s, and spinned-down. Samples (0.006–0.012 ml/well depending on the target protein) were loaded into a pre-cast mini-PROTEAN gel (Bio-Rad), run, and transferred in a min Trans-blot

cell according to manufacturer's instructions and consumables (Bio-Rad). Membranes were probe with primary (AHCY, ProteinTech 10757-2-AP, 1:750; H4R3Me2, ABCAM ab5823, 1:1000; H4, ABCAM ab10158, 1:1000 and Cell Signalling 13919S, 1:1000; Actin, Sigma A5441, 1:5000; CK1D1 and CK1D2, Yanaihara Research Institute, 1:1000) and secondary (anti-rabbit, Amersham NA934, 1:10,000; anti-mouse, Amersham NA931, 1:50,000) antibodies overnight at 4 °C or 1 h at room temperature, respectively, and followed by three washes of 10 min at room temperature with Tris Buffered Saline 0.1% tween20. Proteins were detected using chemiluminescent substrate (Amersham ECL-Prime), pictures of membranes were acquired with a G:Box (Syngene), and integrated intensities quantified with ImageJ available at https://imagej.nih.gov/.

CK1D1 and CK1D2 antibodies were raised by the Yanaihara Research Institute using the peptides KLH-Cys-IPGRVASSGLQSVVHR for CK1D1 and KLH-Cys-NSIPFEHHG for CK1D2, corresponding to the specific C-terminal tail of each isoform. The anti-CK1D2 was affinity purified. The specificity of CK1D1 and CK1D2 antibodies was first ascertained on lysates from PER2::LUC cells transfected with CK1D1 and CK1D2 described previously[56], shown in Fig. 4e. Both antibodies are available on request from Jean-Michel Fustin.

**PRMT5 and AHCY enzymatic assays.** Each respective drug and vehicle was diluted with the assay buffer to the final concentrations indicated in the graphs. Assays were performed following the manufacturers' instructions for PRMT5 (AMS biotechnology 52002L) and AHCY (ABCAM ab204694). Luminescence (PRMT5) and fluorescence (AHCY) were quantified using a BioTek Synergy plate reader (BioTek Instruments Inc.).

**Metabolite quantification by LC-MS/MS.** Cells cultivated in 10 cm Petri dishes (Corning) until confluence were treated with each respective chemical and returned to the incubator for 24 h at 37 °C, 5% CO$_2$. Cells were washed twice with 10 ml 5% mannitol (Sigma-Aldrich), the mannitol was carefully and completely removed before 0.9 ml 100% methanol was added onto the cells, firmly rocking the dish so that the methanol covers the cell monolayer. Dishes were tipped, and 0.6 ml water containing 125 ng/ml BIS-TRIS (Sigma-Aldrich) was added directly into the pool of methanol forming in the corner of the dish before rocking the dish again to cover the cell monolayer. The water/methanol mix was collected from the corner of the tipped dish and transferred to a 1.5 ml microtube. Tubes were kept at room temperature until all dishes were processed, randomly. Samples were centrifuged at 20,000 × g, 4 °C for 30 min, the supernatant transferred to a new tube, centrifuged again at 20,000 × g, 4 °C for 10 min, and the final supernatant transferred to a new 1.5 ml microtube.

Prior to analysis, 200 µl of sample was dried in a centrifugal vacuum concentrator and resuspended in 100 µl acetonitrile and water in a ratio of 5:1. The sample was centrifuged at 20,000 × g for 3 min and the top 80 µl was transferred to a glass autosampler vial with 300 µl insert and capped.

Liquid chromatography-mass spectrometry analysis was performed using a Thermo-Fisher Ultimate 3000 HPLC system consisting of an HPG-3400RS high-pressure gradient pump, TCC 3000 SD column compartment, and WPS 3000 Autosampler, coupled to a SCIEX 6600 TripleTOF Q-TOF mass spectrometer with TurboV ion source. The system was controlled by SCIEX Analyst 1.7.1, DCMS Link, and Chromeleon Xpress software.

A sample volume of 5 µL was injected by pulled loop onto a 5 µL sample loop with 150 µl post-injection needle wash with 9:1 acetonitrile and water. Injection cycle time was 1 min per sample. Separations were performed using an Agilent Poroshell 120 HILIC-Z PEEK-lined column with dimensions of 150 mm length, 2.1 mm diameter, and 2.7 µm particle size equipped with a guard column of the same phase. Mobile phase A was water with 10 mM ammonium formate and 0.1% formic acid, mobile phase B was 9:1 acetonitrile and water with 10 mM ammonium formate and 0.1% formic acid. Separation was performed by gradient chromatography at a flow rate of 0.25 ml/min, starting at 98% B for 3 min, ramping to 5% B over 20 min, hold at 5% B for 1 min, then back to 98% B. Re-equilibration time was 5 min. Total run time including 1 min injection cycle was 30 min.

The mass spectrometer was run in positive mode under the following source conditions: curtain gas pressure, 50 psi; ionspray voltage, 5500 V; temperature, 400 °C; ESI nebulizer gas pressure, 50 psi; heater gas pressure, 70 psi; declustering potential, 80 V.

Data were acquired in a data-independent manner using SWATH in the range of 50–1000 m/z, split across 78 variable-size windows (79 experiments including TOF survey scan), each with an accumulation time of 20 ms. Total cycle time was 1.66 s. Collision energy of each SWATH window was determined using the formula CE (V) = 0.084 × m/z + 12 up to a maximum of 55 V.

Acquired data were processed in MultiQuant 3.0.2. Peaks from MS1 and MS2 data were picked and matched against a metabolite library of 235 standards, based on retention time and mass error of ±0.025 Da. Data exported from MultiQuant 3.0.2 was further sorted, filtered, and scored using a custom VBA macro in Excel, based on presence, peak area, and coelution of precursor and fragment ions.

For quantification from animal tissues, mice were sacrificed by cervical dislocation followed by cessation of blood circulation, and ~50 mg of each tissue was immediately dissected and snap-frozen in liquid nitrogen. Tissues were weighed then homogenized using Qiagen's TissueLyzer in 800 µl ice-cold 50:50 methanol:water solution per 50 mg tissue, with BIS-TRIS (Sigma-Aldrich) at

125 ng/ml was used as an internal control, in 2 ml microtubes containing one 3 mm Tungsten carbide bead (Qiagen), for 10 min at a frequency of 25 Hertz. Samples were then centrifuged at 12,000 × g at 4 °C, the clear supernatant transferred to a new 1.5 ml microtube and kept at −80 °C until analysis, as above.

**Metabolomics.** Cells were cultivated and metabolites extracted for metabolite quantification by HPLC but followed by filtration through a VIVASPIN 500 3KDa cut-off filters (Sartorius AG) then kept at −80 °C until shipped on dry ice to Human Metabolome Technologies (HMT) for analysis. At HMT, filtrates were centrifugally concentrated and resuspended in 0.05 mL of ultrapure water immediately before measurement. Cationic compounds were measured in the cation mode of metabolome analysis based on the methods previously described[73–75] using CE-TOFMS Agilent CE-TOFMS system (Agilent Technologies) equipped with a Fused silica capillary, i.d. 50 µm × 80 cm. Analytical Conditions were: run buffer Cation buffer solution (p/n: H3301-1001); rinse buffer Cation buffer solution (p/n: H3301-1001); sample injection Pressure injection at 50 mbar, 5 s; CE voltage, Positive, 30 kV; MS ionization, ESI, Positive; MS capillary voltage 4000 V; MS scan range m/z 50–1000; Sheath liquid, HMT sheath liquid (p/n: H3301-1020). Analytical conditions for anionic compounds with MS Agilent 6460 TripleQuad LC/MS equipped with Fused silica capillary, i.d. 50 µm × 80 cm were: anion buffer solution (p/n: H3302-1021); rinse buffer anion buffer solution (p/n: H3302-1021); pressure injection at 50 mbar for 25 s; CE voltage, 30 kV; MS ionization, ESI Positive and negative; MS capillary voltage 4000 V for positive and 3500 V for negative; HMT sheath liquid (p/n: I3300-1030).

Peaks detected in CE-TOFMS analysis were extracted using automatic integration software (MasterHands ver.2.17.1.11 developed at Keio University)[76] and those in CE-QqQMS analysis were extracted using automatic integration software (MassHunter Quantitative Analysis B.06.00 Agilent Technologies) in order to obtain peak information including m/z, migration time (MT), and peak area. The peaks were annotated based on the migration times in CE and m/z values determined by TOFMS. Putative metabolites were then assigned from HMT metabolite database on the basis of m/z and MT. The tolerance was ±0.5 min in MT and ±10 ppm in m/z.

In addition, absolute quantification was performed in 116 metabolites including glycolytic and TCA cycle intermediates, amino acids, and nucleic acids. All the metabolite concentrations were calculated by normalizing the peak area of each metabolite with respect to the area of the internal standard and by using standard curves, which were obtained by three-point calibrations.

Metabolome data were analyzed using Metaboanalyst 5.0 (https://www.metaboanalyst.ca/)[77–79], without normalization nor data transformation, by one-way ANOVA followed by Fisher's LSD with an adjusted p-value (FDR) cutoff <0.05. Only known and significantly regulated metabolites were included in the heatmaps in Figs. 2 and 3.

**mRNA methylation quantification.** Total RNA was extracted from 10 cm Petri dishes following Trizol's protocol (Invitrogen). Messenger RNA was purified using NEBNext® Poly(A) mRNA Magnetic Isolation Module (NEB) following manufacturer's protocol, scaled up to 100 µg of total RNA input per sample. An average of 2 µg mRNA per sample was obtained, purity was checked with a Tapestation (Agilent) and mRNA samples were sent to Tamaserv (Germany) for quantitative LC-MS analysis of methylated nucleotides.

**RNASeq.** Total RNA was submitted to the Genomic Technologies Core Facility (GTCF) of the University of Manchester. Quality and integrity of the RNA samples were assessed using a 4200 TapeStation (Agilent Technologies) and then libraries generated using the Illumina® Stranded mRNA Prep. Ligation kit (Illumina, Inc.) according to the manufacturer's protocol. Briefly, total RNA (typically 0.025–1 µg) was used as input material from which polyadenylated mRNA was purified using poly-T, oligo-attached, magnetic beads. Next, the mRNA was fragmented under elevated temperature and then reverse-transcribed into first-strand cDNA using random hexamer primers and in the presence of Actinomycin D (thus improving strand specificity whilst mitigating spurious DNA-dependent synthesis). Following removal of the template RNA, second-strand cDNA was then synthesized to yield blunt-ended, double-stranded cDNA fragments. Strand specificity was maintained by the incorporation of deoxyuridine triphosphate (dUTP) in place of dTTP to quench the second strand during subsequent amplification. Following a single adenine (A) base addition, adapters with a corresponding, complementary thymine (T) overhang were ligated to the cDNA fragments. Pre-index anchors were then ligated to the ends of the double-stranded cDNA fragments to prepare them for dual indexing. A subsequent PCR amplification step was then used to add the index adapter sequences to create the final cDNA library. The adapter indices enabled the multiplexing of the libraries, which were pooled prior to cluster generation using a cBot instrument. The loaded flow-cell was then paired-end sequenced (76 + 76 cycles, plus indices) on an Illumina HiSeq4000 instrument. Finally, the output data was demultiplexed and BCL-to-Fastq conversion performed using Illumina's bcl2fastq software, version 2.20.0.422.

Raw reads were uploaded to the Galaxy server[80] and its instance at the University of Manchester (https://centaurus.itservices.manchester.ac.uk/), QCed and trimmed using Trimmomatic[81], then aligned to the mouse genome GRCm39

guided by the vM27.annotations from GENCODE[82] using HISAT2 aligner[83]. Transcripts were assembled and quantified using Stringtie[84,85], and differential expression analysis was performed using edgeR on genes with a CPM of at least 1 in all samples[86,87]. Hierarchical clustering and heatmap were performed on GenePettern[88]. Raw data and associated files are available on NCBI's Gene Expression Omnibus repository, accession number GSE184525. GOEnrichment (https://github.com/DanFaria/GOEnrichment) was performed on the Galaxy server.

**Statistics and reproducibility**. All statistics were calculated using GraphPad Prism version 9 with appropriate tests described in the figure legends. The experiments shown in Figs. 1, 3h, 3i, 4b, 4d, 4e, 4f, 4g, S1, S2, S3, S4, S5, S6 have been performed at least three times with consistent results. Data for all replicates are available on request.

**Reporting summary**. Further information on research design is available in the Nature Research Reporting Summary linked to this article.

## Data availability

RNAseq data and associated files have been deposited in NCBI's Gene Expression Omnibus repository, accession number GSE184525. Uncropped blot membranes and actograms for all animals are provided as Supplementary Fig. 7 and 8, respectively.

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

## Acknowledgements

This work was supported by the Medical Research Council (Future Leaders Fellowship MR/S031812/1), by the Ministry of Education, Culture, Sports, Science and Technology of Japan (Grant-in-aid for Scientific Research on Innovative Areas 26116713 (J.-M.F.); Grant-in-aid for Young Scientists 26870283 (J.-M.F.); Grant-in-aid for Scientific Research A 18H04015 (H.O.)), and by a grant for Core Research for Evolutional Science and Technology, Japan Science and Technology Agency CREST/ JPMJCR14W3 (H.O.). J.-M.F was also supported by grants from the Kato Memorial Bioscience Foundation, the Senri Life Science Foundation (S-26003), the Mochida Memorial Foundation for Medical and Pharmaceutical Research, and the Kyoto University internal grant ISHIZUE. We thank Dr. Carolina Greco for the HA-AHCY vector, and Prof Mark Helm for his advice on methylated nucleotides quantification. The authors would like to acknowledge the help of Dr. Peter Briggs for the use of the local Galaxy service provided by the Bioinformatics Core Facility and IT Services at the University of Manchester.

## Author contributions

J.-M.F. designed the project, performed experiments, and wrote the paper. K.F., K.I., S.Y., M.Y., and Y.T. contributed to the experiments shown in Figs. 1–3. B.S. contributed to the experiments shown in Figs. 3–5. G.T. produced MS data shown in Fig. 4c. A.H. generated raw RNAseq data used in Fig. 5b. H.O. contributed to the design of the project.

## Competing interests

The authors declare no competing interests.
