## [Peer Review File · Communications Biology]

Reviewers' comments:

Reviewer #1 (Remarks to the Author):

This manuscript is very well written and interesting. The authors discovered that excess SAM inhibits, rather than promotes, methylation through its downstream metabolite adenine. Adenine is suggested to play a role in methylation as an inhibitor of AHCY. Indeed, AHCY overexpression rescues adenine-dependent phenotype. Although the mechanism is not understood, the decreased methylation led to lengthening circadian rhythm in cells. Finally, the authors provide the data showing this is also observed in mice.

This discovery per se is of great importance, which gives a striking caution to society as SAM is indeed widely used as a supplement. Scientifically, SAM-dependent inhibition of methylation is an unexpected finding. Careful metabolite analysis provided sufficient evidence to conclude adenine is a key metabolite mediating the effect of SAM. This reviewer has come up with several experiments to improve the manuscript, the two of which are shown below, but they might not be mandatory to publish. I can support the publication as it stands.

1, To confirm whether adenine is the responsible metabolite, the authors may test an inhibition of MTAP (by RNAi/inhibitor) to rescue the SAM/MTA supplementation-induced phenotypes (decreases methylation and prolongs circadian rhythms). This data would be helpful to discuss whether the effector of excess SAM is only adenine or not. As the authors stated, MTA can directly inhibit methylation (independently of Ahcy) and contribute to SAM-dependent phenotypes.

2, The in vivo data is very interesting. The potential readers of the manuscript would like to know whether excess SAM also increases MTA/adenine in mice.

Reviewer #2 (Remarks to the Author):

Fukumoto et al discussed an interesting topic on excessive intake of a key nutrient, S-Adenosylmethionine. The authors provided in vitro and in vivo evidence suggesting a link between exogenous S-Adenosylmethionine supplement and circadian rhythms. It is an interesting study overall. Few things should be addressed to strengthen the study.

Major:

1) Does exogenous SAM affect cell proliferation/viability?

Does SAM affect circadian rhythms by affecting cell proliferation/viability?

2) What is dietary SAM disposition in vivo? Does dietary SAM affect adenine in mouse blood, or specific tissues?

Minor:

1) Figure legend can be improved to make it more clear. For example what "a", "b", "c" on top of bar graph means, significance level or what?

2) The scale bar on cell photos in FigS1 is hard to read. Use a better resolution.

Reviewer #3 (Remarks to the Author):

Brief summary,

Fukumoto et al. used a circadian reporting system to show methylation inhibition upon excess SAM treatment, and characterized carefully that the mechanism is via SAM catabolism into adenine (and maybe MTA). SAM is the most well-known methylation donor in cells, and this work shows that, counterintuitively, excessive SAM decreases methylation potential (SAM/SAH ratio). Such metabolic mechanism may influence the thinking in the field of using SAM to manipulate methylation state in cells. The reviewer finds a few things that could be further improved.

Abstract:

The reviewer finds the abstract can benefit from a more concise expression.

Intro:

A summary of current research on methylation changes with SAM treatment is needed. Do most people observe SAM increase, or decrease methylation? Or do people "assume" increased methylation in their research? This will strengthen the claim and the necessity of this work. Even if excessive SAM decreases methylation has been reported, but without full characterization of the mechanism, it will be good for the readers to know.

Results:

Does methylation pattern change in other cell lines when fed with excessive SAM? A relevant question is, do mice show methylation pattern change upon SAM food uptake in the circadian cycle experiments? The paper will have a broader impact if excessive SAM decreases methylation can be observed in a more general setting.

Are downstream metabolites of methionine pathway, such as cystathionine and homoserine, relevant for methylation changes? One connection is that serine is important for folate-mediated one-carbon metabolism. This may further clarify whether AHCY inhibition is the main route for methylation perturbation.

This reviewer thinks the manuscript can be shortened to aid communication of the most important findings by reorganizing Fig 1-3. For example, shorten the methionine part in Fig 1 and the nucleoside imbalance part in Fig 3.

Discussion:

Line numbers in the document will be helpful to cite the manuscript. I copy here: "The potential danger of dietary supplements has recently been flagged in *C. elegans*. N-acetyl cysteine (NAC), a thiol-containing precursor of glutathione (GSH) and one of the most popular antioxidant supplements, was shown to disrupt global gene expression and accelerate aging⁵¹. Similar to the fallacy "more SAM is good for you," GSH is actually synthesized intrinsically from cystathionine in a tightly controlled manner that matches the cellular redox status and promote ROS homeostasis; healthy individuals should avoid unnecessary GSH supplementation and trust their own body to know what's best." Although the reviewer shares similar feelings that daily supplementation needs to be more cautious, the paragraph seems to be less relevant to the main theme.

Overall, the reviewer finds this manuscript is done with good quality and adds to the scientific community.

Reviewers' comments:

Reviewer #1 (Remarks to the Author):

This manuscript is very well written and interesting. The authors discovered that excess SAM inhibits, rather than promotes, methylation through its downstream metabolite adenine. Adenine is suggested to play a role in methylation as an inhibitor of AHCY. Indeed, AHCY overexpression rescues adenine-dependent phenotype. Although the mechanism is not understood, the decreased methylation led to lengthening circadian rhythm in cells. Finally, the authors provide the data showing this is also observed in mice.

This discovery per se is of great importance, which gives a striking caution to society as SAM is indeed widely used as a supplement. Scientifically, SAM-dependent inhibition of methylation is an unexpected finding. Careful metabolite analysis provided sufficient evidence to conclude adenine is a key metabolite mediating the effect of SAM. This reviewer has come up with several experiments to improve the manuscript, the two of which are shown below, but they might not be mandatory to publish. I can support the publication as it stands.

1, To confirm whether adenine is the responsible metabolite, the authors may test an inhibition of MTAP (by RNAi/inhibitor) to rescue the SAM/MTA supplementation-induced phenotypes (decreases methylation and prolongs circadian rhythms). This data would be helpful to discuss whether the effector of excess SAM is only adenine or not. As the authors stated, MTA can directly inhibit methylation (independently of AHCY) and contribute to SAM-dependent phenotypes.

We are now reporting the protective action of 2 methionine salvage inhibitors on the effects of SAM (see related Fig. S4):

Inhibition of the methionine salvage pathway rescues the cells from SAM

Next, to prevent adenine and/or MTA synthesis from SAM, we treated PER2::LUC cells with sardomozide, an inhibitor of SAM decarboxylase⁴⁹ (Fig. 2f), the first and rate-limiting enzyme in the methionine salvage pathway⁵⁰, or MTDIA (Methylthio-DADMe-Immucillin A), an inhibitor of MTAP^{51,52} (Fig. 2f), together with different concentrations of SAM. In a manner consistent with our results so far, sardomozide and to a lesser extent MTDIA protected the cells from the period lengthening effects of SAM (Fig. S4a, b). Without excess SAM, sardomozide shortened the period compared to untreated control (Fig. S4a), indicating that the normal rate of the methionine salvage pathway is important for clock function, while MTDIA weakly lengthened the period (Fig. S4b) probably due to the resulting MTA accumulation⁵¹. Together these data demonstrate that the period lengthening caused by SAM depends on its catabolism to MTA and adenine.

2, The in vivo data is very interesting. The potential readers of the manuscript would like to know whether excess SAM also increases MTA/adenine in mice.

That is an excellent suggestion! In Fig. 6 we are now reporting the results of metabolite quantification in the brain and liver of mice fed the test diets, showing that adenine not only increases in the liver, but also that the ratio SAM/SAH decreases in both the liver and the hypothalamus, in a way similar to what was observed in cell cultures. We added this paragraph in the results, related to the new data: “To gain insights into how SAM intake affects systemic methyl metabolism, we quantified selected metabolites from the liver and hypothalamus of mice fed for three weeks with SAM at 100 and 250 mg/Kg and compared with animals fed with unsupplemented control diet (Fig. 6b, c). As expected from *in vitro* experiments, adenine increased in the liver, indicating the methionine salvage pathway is active in this tissue. Interestingly, the SAM/SAH ratio decreased in both tissues, but for different reasons: in the liver, SAH dramatically increased but SAM did not change, while in the brain both SAM and SAH decreased so that SAH>SAM. This clearly shows that SAM supplementation negatively disrupts systemic methyl metabolism.”

We also added a new section in the methods: “For quantification from animal tissues, mice were sacrificed by cervical dislocation followed by cessation of blood circulation, and ~50mg of each tissues were immediately dissected and snap frozen in liquid nitrogen. Tissues were weighed then homogenized using Qiagen’s TissueLyzer in 800µl ice-cold 50:50 methanol:water solution per 50mg tissue, with BIS-TRIS (Sigma-Aldrich) at 125ng/ml was used as an internal control, in 2 ml microtubes containing one 3mm Tungsten carbide bead (Qiagen), for 10 minutes at a frequency of 25 Hertz. Samples were then centrifuged at 12,000xg at 4°C, the clear supernatant transferred to a new 1.5ml microtube and kept at -80°C until analysis, as above.” Please also refer to the legend of Fig. 6.

Reviewer #2 (Remarks to the Author):

Fukumoto et al discussed an interesting topic on excessive intake of a key nutrient, S-Adenosylmethionine. The authors provided *in vitro* and *in vivo* evidence suggesting a link between exogenous S-Adenosylmethionine supplement and circadian rhythms. It is an interesting study overall. Few things should be addressed to strengthen the study.

Major:

1) Does exogenous SAM cell proliferation/viability?

SAM has been shown to have anti-proliferative and pro-apoptotic effects in cancer cell lines (see our response to reviewer 3). In our experiments, however, we used SAM in contact-inhibited, confluent cell populations that are not actively dividing, and did not observe any obvious toxic effects, just like for DZ, adenine or MTA. This can be seen from the stable baseline in our luminescence recordings (Fig. 1c, d): If cells were dying, there would be a progressive decrease in luminescence down to background levels.

Nevertheless, this comment prompted us to perform additional experiments looking at cell proliferation under SAM treatment in actively proliferating cells. The results, now shown as new Fig. S6a, conclusively demonstrate that SAM is highly toxic to dividing cells. We have added the following headed paragraph to the manuscript:

“SAM is highly toxic to dividing cells

The circadian experiments above were performed with confluent cell populations of PER2::LUC MEFs, which are contact-inhibited. This allows for stable and robust luminescence rhythms to be measured. In these populations, no effects of SAM on the cell cycle or cell death were seen. The anti-proliferative and pro-apoptotic effects of SAM in cancer cells²⁴⁻²⁶ however prompted us to test whether SAM affected the growth of PER2::LUC MEFs. Cells cultivated at low confluence were treated with 1 mM SAM or vehicle, and counted every day for 5 days. While control cells grew exponentially, cells treated with SAM initially showed stunted growth followed by a rapid decline (Fig. S6a). These results further highlight SAM as a toxic metabolite with potential uses as an anti-neoplastic agent.”

Please also refer to the updated methods.

Does SAM affect circadian rhythms by affecting cell proliferation/viability?

This is unlikely, as it is known the cell cycle can be inhibited without changing the period of the clock (O’Neill and Hastings, 2008). However, a treatment that affects the cell cycle can also affect the clock independently, for example with kinase inhibitors or translation inhibitors. As mentioned above however, we did not observe any detrimental effects of SAM (or DZ, adenine or MTA...) on cell viability in our experiments performed with contact-inhibited, confluent cells proliferation, as evidenced by stable luminescence measurements.

2)What is dietary SAM disposition in vivo? Does dietary SAM affect adenine in mouse blood, or specific tissues?

That is an excellent suggestion! In Fig. 6 we are now reporting the results of metabolite quantification in the brain and liver of mice fed the test diets, showing that adenine not only increases in the liver, but also that the ratio SAM/SAH decreases in both the liver and the hypothalamus, in a way similar to what was observed in cell cultures. We added this paragraph in the results, related to the new data: “To gain insights into how SAM intake affects systemic methyl metabolism, we quantified selected metabolites from the liver and hypothalamus of mice fed for three weeks with SAM at 100 and 250 mg/Kg and compared with animals fed with unsupplemented control diet (Fig. 6b, c). As expected from *in vitro* experiments, adenine increased in the liver, indicating the methionine salvage pathway is active in this tissue. Interestingly, the SAM/SAH ratio decreased in both tissues, but for different reasons: in the liver, SAH dramatically increased but SAM did not change, while in the brain both SAM and SAH decreased so that SAH>SAM. This clearly shows that SAM supplementation negatively disrupts systemic methyl metabolism.”

We also added a new section in the methods: “For quantification from animal tissues, mice were sacrificed by cervical dislocation followed by cessation of blood circulation, and ~50mg of each tissues were immediately dissected and snap frozen in liquid nitrogen. Tissues were weighed then homogenized using Qiagen’s TissueLyzer in 800µl ice-cold 50:50 methanol:water solution per 50mg tissue, with BIS-TRIS (Sigma-Aldrich) at 125ng/ml was used as an internal control, in 2 ml microtubes containing one 3mm Tungsten carbide bead (Qiagen), for 10 minutes at a frequency of 25 Hertz. Samples were then centrifuged at 12,000xg at 4°C, the clear supernatant transferred to a new 1.5ml microtube and kept at -80°C until analysis, as above.”

Please also refer to the legend of Fig. 6.

Minor:

1) Figure legend can be improved to make it more clear. For example, what "a", "b", "c" on top of bar graph means, significance level or what?

Yes, this labelling system is often used to indicate significance in complex multiple comparisons, as explained in the figure legends.

2) The scale bar on cell photos in FigS1 is hard to read. Use a better resolution.

Apologies. We have increased the size of the pictures as well as improved the readability of the scale bar.

Reviewer #3 (Remarks to the Author):

Brief summary,

Fukumoto et al. used a circadian reporting system to show methylation inhibition upon excess SAM treatment, and characterized carefully that the mechanism is via SAM catabolism into adenine (and maybe MTA). SAM is the most well-known methylation donor in cells, and this work shows that, counterintuitively, excessive SAM decreases methylation potential (SAM/SAH ratio). Such metabolic mechanism may influence the thinking in the field of using SAM to manipulate methylation state in cells. The reviewer finds a few things that could be further improved.

Abstract:

The reviewer finds the abstract can benefit from a more concise expression.

We have reworked the abstract in a way we hope will satisfy this reviewer: "The global dietary supplement market is valued at over USD 100 billion. One popular dietary supplement, S-adenosylmethionine, is marketed to improve joints, liver health and emotional well-being in the US since 1999, and has been a prescription drug in Europe to treat depression and arthritis since 1975, but recent studies questioned its efficacy. In our body, S-adenosylmethionine is critical for the methylation of nucleic acids, proteins and many other targets. It is believed that more S-adenosylmethionine is better since it would stimulate methylations and improve health.

Previously, we have shown that methylation reactions regulate biological rhythms in many organisms. Here, using biological rhythms to assess the effects of exogenous S-adenosylmethionine, we reveal that excess S-adenosylmethionine disrupts rhythms and, rather than promoting methylation, is catabolized to adenine and methylthioadenosine, toxic methylation inhibitors. These findings further our understanding of methyl metabolism and question the safety of S-adenosylmethionine as a supplement."

Intro:

A summary of current research on methylation changes with SAM treatment is needed. Do most people observe SAM increase, or decrease methylation? Or do people "assume" increased

methylation in their research? This will strengthen the claim and the necessity of this work. Even if excessive SAM decreases methylation has been reported, but without full characterization of the mechanism, it will be good for the readers to know.

This is a great suggestion. The problem with SAM is that it is at the crossroad of three metabolic pathways: methylation, glutathione synthesis and polyamine synthesis. The benefits of SAM as a supplement or treatment are not supported by any clear mechanistic evidence, and clinical tests (already cited) have assumed more SAM would be beneficial, but only looked at the effects, or lack thereof, on various symptoms. In the introduction, we have added/modified the following paragraphs.

The use of SAM for the treatment of depression originates from reports showing that severely depressed patients and patients with Alzheimer's dementia had lower SAM levels in the cerebrospinal fluid¹², highlighting a link between depression and methyl metabolism¹³.

(...)

Importantly, the consequences of chronic oral SAM administration on methyl metabolism are unknown, and the few existing studies have only looked at the immediate effects (<24h) of SAM on human plasma methyl metabolites²².

(...)

Indeed, in contrast with these unsubstantiated benefits of SAM, SAM has clearly been demonstrated to have anti-proliferative, pro-apoptotic and anti-metastatic effects in cancer cell lines, although the mechanisms were not identified²⁴⁻²⁶. While this may be of considerable value in cancer chemotherapy, the induction of cell cycle arrest further indicates SAM may potentially be toxic. Interestingly, in liver cancer cells SAM was shown to cause genome-wide hypomethylation as well as hypermethylation of DNA depending on the locus, indicating that more SAM does not necessarily mean more methylations²⁶.

Results:

Does methylation pattern change in other cell lines when fed with excessive SAM?

We tested SAM on a human osteosarcoma cell line (U-2 OS) and obtained identical effects on circadian rhythms, indicating SAM is toxic also in these cells, but we judged the full complement of metabolome/methylation experiments was not necessary. We have now added this sentence in the results:

"SAM affected a human osteosarcoma cell line (U-2 OS) expressing bioluminescent clock reporter³³ in the same way (Fig. S1d), suggesting a conserved mechanism."

Please also refer to the updated methods.

A relevant question is, do mice show methylation pattern change upon SAM food uptake in the circadian cycle experiments? The paper will have a broader impact if excessive SAM decreases methylation can be observed in a more general setting.

That is an excellent suggestion! In Fig. 6 we are now reporting the results of metabolite quantification in the brain and liver of mice fed the test diets, showing that adenine not only increases in the liver, but also that the ratio SAM/SAH decreases in both the liver and the

hypothalamus, in a way similar to what was observed in cell cultures. We added this paragraph in the results, related to the new data: “To gain insights into how SAM intake affects systemic methyl metabolism, we quantified selected metabolites from the liver and hypothalamus of mice fed for three weeks with SAM at 100 and 250 mg/Kg and compared with animals fed with unsupplemented control diet (Fig. 6b, c). As expected from *in vitro* experiments, adenine increased in the liver, indicating the methionine salvage pathway is active in this tissue. Interestingly, the SAM/SAH ratio decreased in both tissues, but for different reasons: in the liver, SAH dramatically increased but SAM did not change, while in the brain both SAM and SAH decreased so that SAH>SAM. This clearly shows that SAM supplementation negatively disrupts systemic methyl metabolism.”

We also added a new section in the methods: “For quantification from animal tissues, mice were sacrificed by cervical dislocation followed by cessation of blood circulation, and ~50mg of each tissues were immediately dissected and snap frozen in liquid nitrogen. Tissues were weighed then homogenized using Qiagen’s TissueLyzer in 800µl ice-cold 50:50 methanol:water solution per 50mg tissue, with BIS-TRIS (Sigma-Aldrich) at 125ng/ml was used as an internal control, in 2 ml microtubes containing one 3mm Tungsten carbide bead (Qiagen), for 10 minutes at a frequency of 25 Hertz. Samples were then centrifuged at 12,000xg at 4°C, the clear supernatant transferred to a new 1.5ml microtube and kept at -80°C until analysis, as above.” Please also refer to the legend of Fig. 6.

Are downstream metabolites of methionine pathway, such as cystathionine and homoserine, relevant for methylation changes? One connection is that serine is important for folate-mediated one-carbon metabolism. This may further clarify whether AHCY inhibition is the main route for methylation perturbation.

The fact that cystathionine and homoserine change in opposite direction with adenine or SAM (Fig. 3a) despite similar effects of adenine or SAM on circadian rhythms, together with the lack of effects of homocysteine and SAH on circadian rhythms, indicate that cystathionine and homoserine are not relevant in this study.

Serine is indeed linked to 1C metabolism. However, in our metabolome data (Fig. 3a), SAM dramatically increase serine in cell cultures (>5-fold), while only a mild increase (<2-fold) can be observed with adenine, despite similar effects of adenine or SAM on the period. Why serine increases? Via mitochondrial folate metabolism, serine is a major 1-C donor for cytoplasmic reactions (references in Yang and Voulsen, 2016), suggesting that the carbon of methionine from homocysteine remethylation originates mainly from serine. When SAM is in excess (or increase under adenine treatment), allosteric inactivation of MTHFR by SAM will likely cause the accumulation of CH₂-THF in the cytoplasm that will then be used by SHMT to synthesise more serine. Since we have no data on folates, and since the only consistent effect of adenine and SAM on metabolites is the quantitatively similar increase in SAH, such a discussion about an unlikely role of serine would be out of the scope of this manuscript, however.

This reviewer thinks the manuscript can be shortened to aid communication of the most important findings by reorganizing Fig 1-3. For example, shorten the methionine part in Fig 1 and the nucleoside imbalance part in Fig 3.

We respectfully disagree. We do believe it is important to provide comparison with the effects of other methyl cycle metabolites (Fig. 1), and to first disprove alternative hypotheses that could explain our results (Fig 3).

Discussion:

Line numbers in the document will be helpful to cite the manuscript. I copy here: “The potential danger of dietary supplements has recently been flagged in *C. elegans*. N-acetyl cysteine (NAC), a thiol-containing precursor of glutathione (GSH) and one of the most popular antioxidant supplements, was shown to disrupt global gene expression and accelerate aging⁵¹. Similar to the fallacy “more SAM is good for you,” GSH is actually synthesized intrinsically from cystathionine in a tightly controlled manner that matches the cellular redox status and promote ROS homeostasis; healthy individuals should avoid unnecessary GSH supplementation and trust their own body to know what’s best.” Although the reviewer shares similar feelings that daily supplementation needs to be more cautious, the paragraph seems to be less relevant to the main theme.

We agree with this comment. We have now removed this paragraph.

Overall, the reviewer finds this manuscript is done with good quality and adds to the scientific community.

Reviewers' comments:

Reviewer #1 (Remarks to the Author):

The authors have nicely added the metabolite analysis in mice. This data improved the study massively and increased the impact. I can now fully support the publication.

Reviewer #2 (Remarks to the Author):

Comments have been addressed. It is suitable for publication.

Reviewer #3 (Remarks to the Author):

Overall, the authors address most of the points raised previously except a few minor ones.

1. Intro; The response "The benefits of SAM as a supplement or treatment are not supported by any clear mechanistic evidence, and clinical tests (already cited) have assumed more SAM would be beneficial, but only looked at the effects, or lack thereof, on various symptoms" seems to be not compatible with the abstract "It is believed that more S-adenosylmethionine is better since it would stimulate methylations and improve health". Which one is more accurate?

2. Results: New data are appreciated. In the legend, "All graph data shown are mean +/- SEM of 2 (a, b) or 4 (d) culture wells", and in the panel bar chart there are 4 data points for all bars. Echoing with Reviewer 2, the figure legend could benefit from better explaining the a,b,c,d.

3. Results: The reviewer believes that the authors can decide how to present their work, but would still suggest the authors to move some control experiments to the supplementary figures (for example, the Polyamines section is good), so that the main figures are more focused.

Reviewers' comments:

Reviewer #1 (Remarks to the Author):

This manuscript is very well written and interesting. The authors discovered that excess SAM inhibits, rather than promotes, methylation through its downstream metabolite adenine. Adenine is suggested to play a role in methylation as an inhibitor of AHCY. Indeed, AHCY overexpression rescues adenine-dependent phenotype. Although the mechanism is not understood, the decreased methylation led to lengthening circadian rhythm in cells. Finally, the authors provide the data showing this is also observed in mice.

This discovery per se is of great importance, which gives a striking caution to society as SAM is indeed widely used as a supplement. Scientifically, SAM-dependent inhibition of methylation is an unexpected finding. Careful metabolite analysis provided sufficient evidence to conclude adenine is a key metabolite mediating the effect of SAM. This reviewer has come up with several experiments to improve the manuscript, the two of which are shown below, but they might not be mandatory to publish. I can support the publication as it stands.

1, To confirm whether adenine is the responsible metabolite, the authors may test an inhibition of MTAP (by RNAi/inhibitor) to rescue the SAM/MTA supplementation-induced phenotypes (decreases methylation and prolongs circadian rhythms). This data would be helpful to discuss whether the effector of excess SAM is only adenine or not. As the authors stated, MTA can directly inhibit methylation (independently of AHCY) and contribute to SAM-dependent phenotypes.

We are now reporting the protective action of 2 methionine salvage inhibitors on the effects of SAM (see related Fig. S4):

Inhibition of the methionine salvage pathway rescues the cells from SAM

Next, to prevent adenine and/or MTA synthesis from SAM, we treated PER2::LUC cells with sardomozide, an inhibitor of SAM decarboxylase⁴⁹ (Fig. 2f), the first and rate-limiting enzyme in the methionine salvage pathway⁵⁰, or MTDIA (Methylthio-DADMe-Immucillin A), an inhibitor of MTAP^{51,52} (Fig. 2f), together with different concentrations of SAM. In a manner consistent with our results so far, sardomozide and to a lesser extent MTDIA protected the cells from the period lengthening effects of SAM (Fig. S4a, b). Without excess SAM, sardomozide shortened the period compared to untreated control (Fig. S4a), indicating that the normal rate of the methionine salvage pathway is important for clock function, while MTDIA weakly lengthened the period (Fig. S4b) probably due to the resulting MTA accumulation⁵¹. Together these data demonstrate that the period lengthening caused by SAM depends on its catabolism to MTA and adenine.

2, The in vivo data is very interesting. The potential readers of the manuscript would like to know whether excess SAM also increases MTA/adenine in mice.

That is an excellent suggestion! In Fig. 6 we are now reporting the results of metabolite quantification in the brain and liver of mice fed the test diets, showing that adenine not only increases in the liver, but also that the ratio SAM/SAH decreases in both the liver and the hypothalamus, in a way similar to what was observed in cell cultures. We added this paragraph in the results, related to the new data: “To gain insights into how SAM intake affects systemic methyl metabolism, we quantified selected metabolites from the liver and hypothalamus of mice fed for three weeks with SAM at 100 and 250 mg/Kg and compared with animals fed with unsupplemented control diet (Fig. 6b, c). As expected from *in vitro* experiments, adenine increased in the liver, indicating the methionine salvage pathway is active in this tissue. Interestingly, the SAM/SAH ratio decreased in both tissues, but for different reasons: in the liver, SAH dramatically increased but SAM did not change, while in the brain both SAM and SAH decreased so that SAH>SAM. This clearly shows that SAM supplementation negatively disrupts systemic methyl metabolism.”

We also added a new section in the methods: “For quantification from animal tissues, mice were sacrificed by cervical dislocation followed by cessation of blood circulation, and ~50mg of each tissues were immediately dissected and snap frozen in liquid nitrogen. Tissues were weighed then homogenized using Qiagen’s TissueLyzer in 800µl ice-cold 50:50 methanol:water solution per 50mg tissue, with BIS-TRIS (Sigma-Aldrich) at 125ng/ml was used as an internal control, in 2 ml microtubes containing one 3mm Tungsten carbide bead (Qiagen), for 10 minutes at a frequency of 25 Hertz. Samples were then centrifuged at 12,000xg at 4°C, the clear supernatant transferred to a new 1.5ml microtube and kept at -80°C until analysis, as above.” Please also refer to the legend of Fig. 6.

Reviewer #2 (Remarks to the Author):

Fukumoto et al discussed an interesting topic on excessive intake of a key nutrient, S-Adenosylmethionine. The authors provided *in vitro* and *in vivo* evidence suggesting a link between exogenous S-Adenosylmethionine supplement and circadian rhythms. It is an interesting study overall. Few things should be addressed to strengthen the study.

Major:

1) Does exogenous SAM cell proliferation/viability?

SAM has been shown to have anti-proliferative and pro-apoptotic effects in cancer cell lines (see our response to reviewer 3). In our experiments, however, we used SAM in contact-inhibited, confluent cell populations that are not actively dividing, and did not observe any obvious toxic effects, just like for DZ, adenine or MTA. This can be seen from the stable baseline in our luminescence recordings (Fig. 1c, d): If cells were dying, there would be a progressive decrease in luminescence down to background levels.

Nevertheless, this comment prompted us to perform additional experiments looking at cell proliferation under SAM treatment in actively proliferating cells. The results, now shown as new Fig. S6a, conclusively demonstrate that SAM is highly toxic to dividing cells. We have added the following headed paragraph to the manuscript:

“SAM is highly toxic to dividing cells

The circadian experiments above were performed with confluent cell populations of PER2::LUC MEFs, which are contact-inhibited. This allows for stable and robust luminescence rhythms to be measured. In these populations, no effects of SAM on the cell cycle or cell death were seen. The anti-proliferative and pro-apoptotic effects of SAM in cancer cells²⁴⁻²⁶ however prompted us to test whether SAM affected the growth of PER2::LUC MEFs. Cells cultivated at low confluence were treated with 1 mM SAM or vehicle, and counted every day for 5 days. While control cells grew exponentially, cells treated with SAM initially showed stunted growth followed by a rapid decline (Fig. S6a). These results further highlight SAM as a toxic metabolite with potential uses as an anti-neoplastic agent.”

Please also refer to the updated methods.

Does SAM affect circadian rhythms by affecting cell proliferation/viability?

This is unlikely, as it is known the cell cycle can be inhibited without changing the period of the clock (O’Neill and Hastings, 2008). However, a treatment that affects the cell cycle can also affect the clock independently, for example with kinase inhibitors or translation inhibitors. As mentioned above however, we did not observe any detrimental effects of SAM (or DZ, adenine or MTA...) on cell viability in our experiments performed with contact-inhibited, confluent cells proliferation, as evidenced by stable luminescence measurements.

2)What is dietary SAM disposition in vivo? Does dietary SAM affect adenine in mouse blood, or specific tissues?

That is an excellent suggestion! In Fig. 6 we are now reporting the results of metabolite quantification in the brain and liver of mice fed the test diets, showing that adenine not only increases in the liver, but also that the ratio SAM/SAH decreases in both the liver and the hypothalamus, in a way similar to what was observed in cell cultures. We added this paragraph in the results, related to the new data: “To gain insights into how SAM intake affects systemic methyl metabolism, we quantified selected metabolites from the liver and hypothalamus of mice fed for three weeks with SAM at 100 and 250 mg/Kg and compared with animals fed with unsupplemented control diet (Fig. 6b, c). As expected from *in vitro* experiments, adenine increased in the liver, indicating the methionine salvage pathway is active in this tissue. Interestingly, the SAM/SAH ratio decreased in both tissues, but for different reasons: in the liver, SAH dramatically increased but SAM did not change, while in the brain both SAM and SAH decreased so that SAH>SAM. This clearly shows that SAM supplementation negatively disrupts systemic methyl metabolism.”

We also added a new section in the methods: “For quantification from animal tissues, mice were sacrificed by cervical dislocation followed by cessation of blood circulation, and ~50mg of each tissues were immediately dissected and snap frozen in liquid nitrogen. Tissues were weighed then homogenized using Qiagen’s TissueLyzer in 800µl ice-cold 50:50 methanol:water solution per 50mg tissue, with BIS-TRIS (Sigma-Aldrich) at 125ng/ml was used as an internal control, in 2 ml microtubes containing one 3mm Tungsten carbide bead (Qiagen), for 10 minutes at a frequency of 25 Hertz. Samples were then centrifuged at 12,000xg at 4°C, the clear supernatant transferred to a new 1.5ml microtube and kept at -80°C until analysis, as above.”

Please also refer to the legend of Fig. 6.

Minor:

1) Figure legend can be improved to make it more clear. For example, what "a", "b", "c" on top of bar graph means, significance level or what?

Yes, this labelling system is often used to indicate significance in complex multiple comparisons, as explained in the figure legends.

2) The scale bar on cell photos in FigS1 is hard to read. Use a better resolution.

Apologies. We have increased the size of the pictures as well as improved the readability of the scale bar.

Reviewer #3 (Remarks to the Author):

Thank you very much again for your additional comments. Please find our response below, coloured as previously.

Brief summary,

Fukumoto et al. used a circadian reporting system to show methylation inhibition upon excess SAM treatment, and characterized carefully that the mechanism is via SAM catabolism into adenine (and maybe MTA). SAM is the most well-known methylation donor in cells, and this work shows that, counterintuitively, excessive SAM decreases methylation potential (SAM/SAH ratio). Such metabolic mechanism may influence the thinking in the field of using SAM to manipulate methylation state in cells. The reviewer finds a few things that could be further improved.

Abstract:

The reviewer finds the abstract can benefit from a more concise expression.

We have reworked the abstract in a way we hope will satisfy this reviewer: "The global dietary supplement market is valued at over USD 100 billion. One popular dietary supplement, S-adenosylmethionine, is marketed to improve joints, liver health and emotional well-being in the US since 1999, and has been a prescription drug in Europe to treat depression and arthritis since 1975, but recent studies questioned its efficacy. In our body, S-adenosylmethionine is critical for the methylation of nucleic acids, proteins and many other targets. It is believed that more S-adenosylmethionine is better since it would stimulate methylations and improve health.

Previously, we have shown that methylation reactions regulate biological rhythms in many organisms. Here, using biological rhythms to assess the effects of exogenous S-adenosylmethionine, we reveal that excess S-adenosylmethionine disrupts rhythms and, rather than promoting methylation, is catabolized to adenine and methylthioadenosine, toxic methylation inhibitors. These findings further our understanding of methyl metabolism and question the safety of S-adenosylmethionine as a supplement."

Intro:

A summary of current research on methylation changes with SAM treatment is needed. Do most

people observe SAM increase, or decrease methylation? Or do people “assume” increased methylation in their research? This will strengthen the claim and the necessity of this work. Even if excessive SAM decreases methylation has been reported, but without full characterization of the mechanism, it will be good for the readers to know.

This is a great suggestion. The problem with SAM is that it is at the crossroad of three metabolic pathways: methylation, glutathione synthesis and polyamine synthesis. The benefits of SAM as a supplement or treatment are not supported by any clear mechanistic evidence, and clinical tests (already cited) have assumed more SAM would be beneficial, but only looked at the effects, or lack thereof, on various symptoms. In the introduction, we have added/modified the following paragraphs.

The use of SAM for the treatment of depression originates from reports showing that severely depressed patients and patients with Alzheimer’s dementia had lower SAM levels in the cerebrospinal fluid¹², highlighting a link between depression and methyl metabolism¹³.

(...)

Importantly, the consequences of chronic oral SAM administration on methyl metabolism are unknown, and the few existing studies have only looked at the immediate effects (<24h) of SAM on human plasma methyl metabolites²².

(...)

Indeed, in contrast with these unsubstantiated benefits of SAM, SAM has clearly been demonstrated to have anti- proliferative, pro- apoptotic and anti- metastatic effects in cancer cell lines, although the mechanisms were not identified²⁴⁻²⁶. While this may be of considerable value in cancer chemotherapy, the induction of cell cycle arrest further indicates SAM may potentially be toxic. Interestingly, in liver cancer cells SAM was shown to cause genome-wide hypomethylation as well as hypermethylation of DNA depending on the locus, indicating that more SAM does not necessarily mean more methylations²⁶.

The response “The benefits of SAM as a supplement or treatment are not supported by any clear mechanistic evidence, and clinical tests (already cited) have assumed more SAM would be beneficial, but only looked at the effects, or lack thereof, on various symptoms” seems to be not compatible with the abstract “It is believed that more S-adenosylmethionine is better since it would stimulate methylations and improve health”. Which one is more accurate?

Both are compatible. The marketing of SAM implies more SAM will increase methylation, but in the scientific literature there are no such evidence. We have clarified this in the abstract to avoid confusion: “The marketing of SAM implies that more S-adenosylmethionine is better since it would stimulate methylations and improve health.”

Results:

Does methylation pattern change in other cell lines when fed with excessive SAM?

We tested SAM on a human osteosarcoma cell line (U-2 OS) and obtained identical effects on circadian rhythms, indicating SAM is toxic also in these cells, but we judged the full complement of metabolome/methylation experiments was not necessary. We have now added this sentence in the results:

“SAM affected a human osteosarcoma cell line (U-2 OS) expressing bioluminescent clock reporter³³ in the same way (Fig. S1d), suggesting a conserved mechanism.”

Please also refer to the updated methods.

New data are appreciated. In the legend, “All graph data shown are mean +/- SEM of 2 (a, b) or 4 (d) culture wells”, and in the panel bar chart there are 4 data points for all bars. Echoing with Reviewer 2, the figure legend could benefit from better explaining the a,b,c,d.

The bar charts of Fig S1a and b show 2 data points for each bar, but the bar chart in Fig S1d show 4 data points, as written in the legend. To avoid confusion, the statement (labels at the top of each bar) was added in all legends whenever the a vs. b (...) occurs. In addition, we have modified the legend of Fig S1 to state: All bar graph data shown are mean +/- SEM of 2 (panel a, b) or 4 (panel d) culture wells.

A relevant question is, do mice show methylation pattern change upon SAM food uptake in the circadian cycle experiments? The paper will have a broader impact if excessive SAM decreases methylation can be observed in a more general setting.

That is an excellent suggestion! In Fig. 6 we are now reporting the results of metabolite quantification in the brain and liver of mice fed the test diets, showing that adenine not only increases in the liver, but also that the ratio SAM/SAH decreases in both the liver and the hypothalamus, in a way similar to what was observed in cell cultures. We added this paragraph in the results, related to the new data: “To gain insights into how SAM intake affects systemic methyl metabolism, we quantified selected metabolites from the liver and hypothalamus of mice fed for three weeks with SAM at 100 and 250 mg/Kg and compared with animals fed with unsupplemented control diet (Fig. 6b, c). As expected from *in vitro* experiments, adenine increased in the liver, indicating the methionine salvage pathway is active in this tissue. Interestingly, the SAM/SAH ratio decreased in both tissues, but for different reasons: in the liver, SAH dramatically increased but SAM did not change, while in the brain both SAM and SAH decreased so that SAH>SAM. This clearly shows that SAM supplementation negatively disrupts systemic methyl metabolism.”

We also added a new section in the methods: “For quantification from animal tissues, mice were sacrificed by cervical dislocation followed by cessation of blood circulation, and ~50mg of each tissues were immediately dissected and snap frozen in liquid nitrogen. Tissues were weighed then homogenized using Qiagen’s TissueLyzer in 800µl ice-cold 50:50 methanol:water solution per 50mg tissue, with BIS-TRIS (Sigma-Aldrich) at 125ng/ml was used as an internal control, in 2 ml microtubes containing one 3mm Tungsten carbide bead (Qiagen), for 10 minutes at a frequency of 25 Hertz. Samples were then centrifuged at 12,000xg at 4°C, the clear supernatant transferred to a new 1.5ml microtube and kept at -80°C until analysis, as above.” Please also refer to the legend of Fig. 6.

Are downstream metabolites of methionine pathway, such as cystathionine and homoserine, relevant for methylation changes? One connection is that serine is important for folate-mediated one-carbon metabolism. This may further clarify whether AHCY inhibition is the main route for methylation perturbation.

The fact that cystathionine and homoserine change in opposite direction with adenine or SAM (Fig. 3a) despite similar effects of adenine or SAM on circadian rhythms, together with the lack of effects of homocysteine and SAH on circadian rhythms, indicate that cystathionine and homoserine are not relevant in this study.

Serine is indeed linked to 1C metabolism. However, in our metabolome data (Fig. 3a), SAM dramatically increase serine in cell cultures (>5-fold), while only a mild increase (<2-fold) can be observed with adenine, despite similar effects of adenine or SAM on the period. Why serine increases? Via mitochondrial folate metabolism, serine is a major 1-C donor for cytoplasmic reactions (references in Yang and Voulsen, 2016), suggesting that the carbon of methionine from homocysteine remethylation originates mainly from serine. When SAM is in excess (or increase under adenine treatment), allosteric inactivation of MTHFR by SAM will likely cause the accumulation of CH₂-THF in the cytoplasm that will then be used by SHMT to synthesise more serine. Since we have no data on folates, and since the only consistent effect of adenine and SAM on metabolites is the quantitatively similar increase in SAH, such a discussion about an unlikely role of serine would be out of the scope of this manuscript, however.

This reviewer thinks the manuscript can be shortened to aid communication of the most important findings by reorganizing Fig 1-3. For example, shorten the methionine part in Fig 1 and the nucleoside imbalance part in Fig 3.

We respectfully disagree. We do believe it is important to provide comparison with the effects of other methyl cycle metabolites (Fig. 1), and to first disprove alternative hypotheses that could explain our results (Fig 3).

The reviewer would still suggest the authors to move some control experiments to the supplementary figures (such as the Polyamines section), so that the main figures are more focused. Either way, the scientific discovery of the paper is solid.

This rearrangement of data has only been suggested by one of the three reviewers and, at this stage of the reviewing process, we are therefore reluctant to make these modifications, as we believe that, although negative, these data are nonetheless important.

Discussion:

Line numbers in the document will be helpful to cite the manuscript. I copy here: "The potential danger of dietary supplements has recently been flagged in *C. elegans*. N-acetyl cysteine (NAC), a thiol-containing precursor of glutathione (GSH) and one of the most popular antioxidant supplements, was shown to disrupt global gene expression and accelerate aging⁵¹. Similar to the fallacy "more SAM is good for you," GSH is actually synthesized intrinsically from cystathionine in a tightly controlled manner that matches the cellular redox status and promote ROS homeostasis; healthy individuals should avoid unnecessary GSH supplementation and trust their own body to know what's best." Although the reviewer shares similar feelings that daily supplementation needs to be more cautious, the paragraph seems to be less relevant to the main theme.

We agree with this comment. We have now removed this paragraph.

Overall, the reviewer finds this manuscript is done with good quality and adds to the scientific community.